# EXPLAINING EMERGENT IN-CONTEXT LEARNING AS KERNEL REGRESSION

## ABSTRACT

Large language models (LLMs) have initiated a paradigm shift in transfer learning. In contrast to the classic pretraining-then-finetuning procedure, in order to use LLMs for downstream prediction tasks, one only needs to provide a few demonstrations, known as in-context examples, without adding more or updating existing model parameters. This in-context learning (ICL) capability of LLMs is intriguing, and it is not yet fully understood how pretrained LLMs acquire such capabilities. In this paper, we investigate the reason why a transformer-based language model can accomplish in-context learning after pre-training on a general language corpus by proposing a kernel-regression perspective of undertanding LLMs' ICL bahaviors when faced with in-context examples. More concretely, we first prove that Bayesian inference on in-context prompts can be asymptotically understood as kernel regression $\hat{y} = \sum_i y_i K(x, x_i) / \sum_i K(x, x_i)$ as the number of in-context demonstrations grows. Then, we empirically investigate the in-context behaviors of language models. We find that during ICL, the attention and hidden features in LLMs match the behaviors of a kernel regression. Finally, our theory provides insights into multiple phenomena observed in the ICL field: why retrieving demonstrative samples similar to test samples can help, why ICL performance is sensitive to the output formats, and why ICL accuracy benefits from selecting in-distribution and representative samples.

## 1 INTRODUCTION

Pre-trained large language models (LLMs) have emerged as powerful tools in the field of natural language processing, demonstrating remarkable performance across a broad range of applications (Wei et al., 2022a; Kojima et al., 2023; Wei et al., 2022b; Brown et al., 2020; Li et al., 2023). They have been used to tackle diverse tasks such as text summarization, sentiment analysis, schema induction and translation, among others (Brown et al., 2020; Radford et al., 2023; Li et al., 2023). One of the most fascinating capabilities of LLMs is their ability to perform in-context learning (ICL), a process in which a language model can make predictions on a test sample based on a few demonstrative examples provided in the input context (Logan IV et al., 2022). This feature makes LLMs particularly versatile and adaptive to different tasks. Studies have found ICL to emerge especially when the size of LLM is large enough and pre-trained over a massive corpus (Wei et al., 2023).

Although intuitive for human learners, ICL poses a mystery for optimization theories because of the significant format shift between ICL prompts and pre-training corpus. There have been lots of efforts to provide a theoretical understanding of how LLMs implement ICL. Some work (Xie et al., 2022; Wang et al., 2023) approaches this problem from a data perspective: they claim that ICL is possible if a model masters Bayesian inference on pre-training distribution. However, they fail to explain how such inference is feasibly implemented in practical language models, nor does their theory provide insights into ICL behaviors. Another stream of work conjectures that under a simple linear setting: $[x, y]$ where $y = w^\top x$ and the input sequence $x$ only has length 1, they can construct a Transformer (Vaswani et al., 2017) to implement gradient descent (GD) algorithm over ICL prompt (Akyürek et al., 2023; von Oswald et al., 2022; Garg et al., 2023). However, this constrained setting diverges from the most interesting part of ICL, as state-of-the-art LLMs work with linguistic tasks where the sequential textual inputs have complex semantic structures, and ICL emerges from pre-training on general-purpose corpus instead of explicit ICL training.

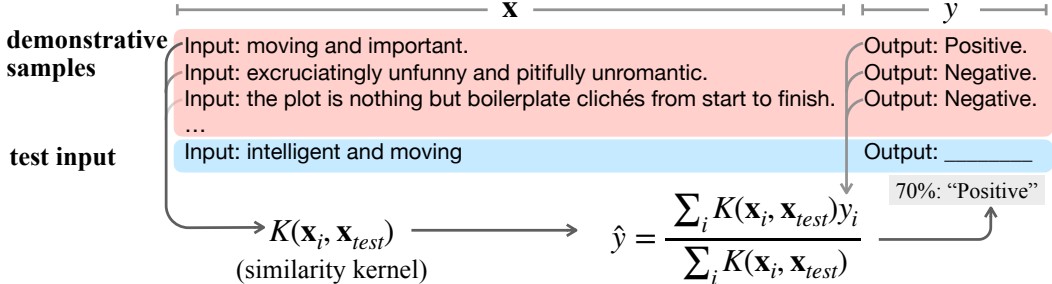

Figure 1: Our results suggests that LLMs might be conducting kernel regression on ICL prompts.

In this work, we delve deeper into the question of *how to understand the mechanism that enables Transformer-based pre-trained LLMs to accomplish in-context learning on sequential data*. We specifically explore the hypothesis that LLMs employ a kernel regression algorithm when confronted with in-context prompts. Kernel regression adopts a non-parametric form

$$\hat{y} = \frac{\sum_i y_i K(x, x_i)}{\sum_i K(x, x_i)} \tag{1}$$

when making predictions, where $K(x, x_i)$ is a kernel that measures the similarity between inputs $x$ and $x_i$. In plain words, it estimates the output $\hat{y}$ on $x$ by drawing information from similar other data points $x_i$ and taking a weighted sum on their $y_i$.

We first provide a theoretical analysis demonstrating that Bayesian inference predictions on in-context prompts converge to a kernel regression in Section 4. In Section 4.2, our results also shed light on numerous phenomena observed in previous empirical studies, such as the advantage of retrieving in-context examples that are similar to the test sample, the sensitivity of ICL performance to the output formats, and why using a group of in-distribution and representative samples improves ICL accuracy.

Following our theoretical investigation, in Section 5 we conduct empirical studies to verify our explanation of in-context learning of LLMs in more detail. Our results reveal that during LLM ICL, the attention map used by the last token to predict the next token is allocated in accordance with our explanation. By plugging attention values into our equation, we are also able to reconstruct the model's output with over 80% accuracy. Moreover, we are able to reveal how information necessary to kernel regression is computed in intermediate LLM layers. In conclusion, we make the following contributions to this work:

- We provide a kernel-regression perspective of understanding how LLMs tackle the ICL problem.
- We conduct empirical analysis to verify that the LLM's attention maps and hidden features match our understanding.
- Out theory provides insights and heuristics to multiple phenomena observed in ICL practice by previous studies and also sheds light on new findings, such as the effect of OOD inputs.

## 2 RELATED WORK

### 2.1 IN-CONTEXT LEARNING

As an intriguing property of large language models, in-context learning has attracted high attention in the research community. There have been numerous studies on empirical analysis of in-context learning, including format and effects of in-context samples (Min et al., 2022b;a; Zhao et al., 2021), selection of in-context samples (Liu et al., 2022; Lu et al., 2022), characteristics of in-context learning behaviors (Zhao et al., 2021; Lu et al., 2022; Khashabi et al., 2022; Wei et al., 2023), and the relation between ICL and pre-training dataset (Wu et al., 2022; Chan et al., 2022).

Going deeper, researchers have also been interested in building a theoretical understanding of why ICL works. One branch of studies investigates ICL from a data perspective: (Xie, 2021; Wang

et al., 2023) demonstrate that a good enough Bayesian inference on pre-training data might cause the emergence of ICL ability. However, they fail to explain if such Bayesian inference is computationally feasible in practical language models, as such Bayesian inference involves unbounded depth computational graphs as the number of samples increases. Our study builds on top of some similar assumptions but goes further to explain how ICL can be accomplished with attention-mechanism in Transformers (Vaswani et al., 2017).

Another angle of explanation is analyzing what algorithms might be implemented in LLMs for ICL, which is closely related to our study. Representative studies include (Akyürek et al., 2023; Garg et al., 2023; von Oswald et al., 2022; Dai et al., 2022), with the majority of them proposing gradient descent (GD) algorithm as a promising candidate answer. However, attempts in explicit construction of GD algorithm in Transformers (Akyürek et al., 2023; von Oswald et al., 2022) mostly assume an oversimplified setting of linear tasks with input length equal to 1, and evaluate Transformers after training on a synthetic dataset (including (Akyürek et al., 2023; Garg et al., 2023)). This is different from ICL's main advantage as an emergent ability on language pre-training, and LLMs are able to work on textual data which involves more complex syntactic and semantic structures.

## 2.2 EMERGENT ABILITY OF LLMS

This is a larger topic that in-context learning is also highly related to. (Wei et al., 2022a; Brown et al., 2020; Kojima et al., 2023; Wei et al., 2022b) showed that abilities including reasoning, in-context learning, few-shot learning, instruction understanding, and multilingualism emerge in large language models after pre-training on massive language data. These impressive and mysterious capacities have boosted significant progress in natural language processing as well as artificial intelligence, but still baffle theoretical analysis. In this work, we make a preliminary step towards understanding ICL as a special case of LM capacity emergence.

## 2.3 ASSOCIATING ATTENTION WITH KERNEL REGRESSION

Chen & Li (2023) presents a related view of associating self-attention with kernel ridge regressions. They focus on a different problem and propose to substitute attention mechanisms with a kernel to tackle the uncertainty calibration problem. Another blog, Olsson et al. (2022), also relates attention to kernel regression but actually defines ICL as "decreasing loss at increasing token indices", rather than the more widely-adopted definition of learning from in-context demonstrations as in ours.

## 3 FORMULATION

### 3.1 PRELIMINARIES: HIDDEN MARKOV MODELS

Following the setting of (Xie et al., 2022), we assume that the pre-training corpus can be modeled by a mixture of HMMs. Each HMM corresponds to a certain task $\theta \in \Theta$. Assuming a large finite number of tasks, one can include all task-specific HMMs into one single HMM. In this unified HMM, let $\mathcal{S}$ be the set of states, and $\mathcal{O}$ be the set of observations where $|\mathcal{O}| = m$. At each time step, state $s_t$ randomly emits one observation $o_t$ and then transits to the next state $s_{t+1}$. $p_{\text{pre-train}}$, $P(s_{t+1} = s'|s_t = s)$ and $P(o_t = o|s_t = s)$ denote the pre-training initial distribution, transition distribution and emission distribution respectively. Under an arbitrary ordering of $\mathcal{S}$ and $\mathcal{O}$, we can define the transition matrix $T : T(s, s') = P(s'|s)$, and emission matrix $B : B(s, o) = P(o|s)$, respectively. We also let $\mathbf{o} = (o_0, \cdots)$ be the full observation sequence, and $\mathbf{o}_{[0:l]}$ denote its first $l$ tokens.

### 3.2 IN-CONTEXT LEARNING

In this work, we consider the following formulation of in-context learning (ICL). Let $\Theta$ be the set of tasks. The distribution of sequences generated by each individual task in the HMM together composes the pre-training distribution. Specifically, each task $\theta \in \Theta$ is associated with a distinct initial state $s_\theta \in \mathcal{S}$ with transition rate lower bound $\epsilon_d$, and the set of all such initial states $\mathcal{S}_{\text{start}} = \{s_\theta | \theta \in \Theta\}$ forms the support of $p_{\text{pre-train}}$.

Following the ICL prompt formulation in (Xie et al., 2022), for a test task $\theta^\star$, the in-context learning prompt follows the format:

$$[S_n, \mathbf{x}_{\text{test}}] = [\mathbf{x}_1, y_1, o^{\text{delim}}, \mathbf{x}_2, y_2, o^{\text{delim}}, \cdots, \mathbf{x}_n, y_n, o^{\text{delim}}, \mathbf{x}_{\text{test}}], \tag{2}$$

where the input-output pairs $[\mathbf{x}_i, y_i]$ are i.i.d. demonstrate samples sampled from $\theta^\star$, and $o^{\text{delim}}$ is the delimiter token with emission rate lower bound $\epsilon_r$. We further make some connections between in-context learning and the HMM model. Note that the probability of generating a sequence from the initial distribution $p_0$ can be expressed as follows(Jaeger, 2000):

$$P(\mathbf{o}_{[0:l]}|p_0) = \mathbf{v}_{p_0}^\top \left( \prod_{i=0}^{l-1} \text{diag}(\mathbf{p}_{o_i})T \right) \text{diag}(\mathbf{p}_{o_l})\mathbf{1}, \tag{3}$$

where $\mathbf{p}_o$ is vector of emission probabilities $P(o|s \in \mathcal{S})$ for $o$. We denote the intermediate matrices as one operator $T_{\mathbf{o}_{[0:l-1]}} = \prod_{i=0}^{l-1} \text{diag}(\mathbf{p}_{o_i})T$. We use a matrix $\Sigma_{p,l}$ to denote the covariance between all of its $d^2$ elements of $\text{vec}(T_{\mathbf{o}_{[0:l-1]}})$ when $\mathbf{o}_{[0:l-1]}$ is generated from initial distribution $p$. For each individual task, we also have $\epsilon_\theta = \inf_l \rho(\Sigma_{p_{\text{pre-train}},l}^{-1} - \Sigma_{s_\theta,l}^{-1})$ to quantify the difference between sequences generated by $s_\theta$ and those from pre-training distribution, where $\rho$ denotes the spectral radius of a matrix. Let $\eta = \sup_{\mathbf{o}_{[0:l-1]}} \|T_{\mathbf{o}_{[0:l]}}\|_F$ be the upper bound of $T_{\mathbf{o}_{[0:l]}}$'s Frobenius-norm.

### 3.3 ASSUMPTIONS

We go on and present the assumptions we borrow from Xie et al. (2022).

**Assumption 1.** *(Delimiter Tokens) The delimiter token indicates the start of sampling of new sequences:*
$$P(s_{t+1} = s \mid o^{delim}) > 0 \to s \in \mathcal{S}_{start}$$

**Remark**: this means that the delimiter tokens are indicative enough, such as the start of a new line before the beginning of a paragraph.

**Assumption 2.** *(Task Distinguishability) The Kullback–Leibler divergence (KL divergence) between the first $l$ tokens between two distinct tasks $\theta \neq \theta'$ is lower-bounded by:*

$$\inf_{\theta, \theta'} D_{KL}\left( P(\mathbf{o}_{[0:l]}|\theta') \parallel P(\mathbf{o}_{[0:l]}|\theta)) \right) = \epsilon_{KL} > \ln \frac{1}{\epsilon_r \epsilon_d}.$$

**Remark**: this requires that tasks are distinguishable from each other. As KL-divergence is non-decreasing with length $l$, it suffices to increase the length $l$ to provide sufficient task information.

## 4 THEORETICAL ANALYSIS

### 4.1 EXPLAINING ICL AS KERNEL REGRESSION

Within the framework presented in Section 3, we pose the following result. The basic idea is that, as the number of samples $n$ increases, inference on the in-context learning prompt converges to a kernel-regression form.

**Theorem 1.** *Let us denote a kernel*
$$\mathcal{K}(\mathbf{x}, \mathbf{x}') = \text{vec}(T_\mathbf{x})^\top \Sigma_{p_{pre-train}}^{-1} \text{vec}(T_{\mathbf{x}'}). \tag{4}$$

*Let $\mathbf{e}(y)$ be the one-hot vector for index $y$. Then the difference between the following logit vector in the form of kernel regression:*
$$\hat{\mathbf{y}} = \frac{\sum_{i=1}^n \mathbf{e}(y_i)\mathcal{K}(\mathbf{x}_{test}, \mathbf{x}_i)}{\sum_{i=1}^n \mathcal{K}(\mathbf{x}_{test}, \mathbf{x}_i)} \tag{5}$$

*and the Bayesian posterior converges polynomially with probability $1 - \delta$:*

$$\|\hat{\mathbf{y}} - P(y \mid [S_n, \mathbf{x}_{test}])\|_\infty = \eta^2 \epsilon_\theta + O\left( \sqrt{\frac{1}{n} \ln \frac{4m}{\delta}} \right) \tag{6}$$

Equation 5 can be interpreted as follows: it calculates the semantic similarity between the test input $\mathbf{x}_{\text{test}}$ and each sample $\mathbf{x}_i$ and aggregates their outputs to compute a most likely prediction for the test sample. This is natural to the motivation of ICL: we encourage the LLM to leverage the pattern provided in demonstrative samples and mimic the pattern to predict the test input. Equation 5 is also similar to the form of attention mechanism used in Transformer decoder models:

$$h = \text{softmax}(q^\top K)V^\top = \frac{\sum_i v_i e^{\langle q, k_i \rangle}}{\sum_i e^{\langle q, k_i \rangle}} \tag{7}$$

where $q$ is the query vector corresponding to the last token, $k, K$ are the key vectors and matrix, and $v, V$ are the value vectors and matrix used in the Transformer, respectively. The only difference is that $e^{<q,k_i>}$ is replaced with a dot product in Equation 5, which can be regarded as a kernel trick. We assume that previously hidden layers are responsible for learning the semantic vectors of sample inputs $\text{vec}(T_\mathbf{x})$. We can then make the following loose analogy between our kernel regression explanation (Equation 5) and the attention mechanism (Equation 7):

- Label information $\mathbf{e}(y_i)$ corresponds with the *value* vector $v_i$

- The similarity kernel $\text{vec}(T_\mathbf{x})^\top \Sigma_{p_{\text{pre-train}}}^{-1} \text{vec}(T_{\mathbf{x}'})$ loosely corresponds to the attention value $e^{\langle q, k_i \rangle}$, where:

- the semantic information $\text{vec}(T_{\mathbf{x}_i})$ corresponds to the *key* vectors $k_i$ and *query* vectors $q_i$ for samples $[\mathbf{x}, y]$.

Here we further explain how the kernel is effected by the LLM. Equation 5 measures similarity between $\mathbf{x}$ and $\mathbf{x}'$ on the space of $\text{vec}(T_\mathbf{x})$ and $\text{vec}(T_{\mathbf{x}'})$. This flattened vector of $T_\mathbf{x}$ defines the "belief state" in HMMs, which determines the conditional probability $P(\cdot|\mathbf{x})$ after prefix $\mathbf{x}$. This is the pre-training objective functionality of LLMs. Therefore, if $x_i$ and $x_{test}$ have similar follow-up conditional probabilities under the LLM, their similarity value will be larger, and vice versa.

One might argue that it is also theoretically possible to directly infer the next token in matrix form $p_{\text{pre-train}}^\top T_{[S_n, \mathbf{x}_{\text{test}}]}$. However, this form involves $2n$ consecutive matrix multiplications. When $n$ increases, this is infeasible for a practical Transformer architecture which is composed of a fixed number of layers. In comparison, Equation 5 only requires semantic information for each sample $\mathbf{x}$ to be provided beforehand, and then applies kernel regression (which can be done by one attention layer) to get the answer. Learning to represent $T_\mathbf{x}$ is probable for preceding layers, as it is also used for ordinary inference $P(y \mid \mathbf{x}) = p_{\text{pre-train}}^\top T_\mathbf{x}$. In experiments in Section 5 we demonstrate that this analogy can explain the ICL behaviors of LLMs to an extent.

## 4.2 Insights Provided by the Explanation

Theorem 1 can provide insights into multiple phenomena in the ICL field observed by previous studies. This is helpful for understanding and predicting the behaviors of ICL, and providing heuristics for future development.

**Retrieving Similar Samples** It is empirically observed (Rubin et al., 2022; Liu et al., 2022) that retrieving demonstrative samples $\mathbf{x}_i$ that is similar to the test input $\mathbf{x}_{\text{test}}$ can benefit ICL performance. This phenomenon is understandable from our explanation. Encouraging the selection of similar samples can be understood as limiting the cosine distance between demonstrative samples $\mathbf{x}_i$ and test sample $\mathbf{x}_{\text{test}}$ in sentence embedding space. This is similar to selecting a smaller "bandwidth" in kernel regression and sampling only from a local window, which reduces the bias in kernel regression. Therefore, devising better retrieval techniques for selecting samples, especially those with similar representations as LLMs, is a promising direction for further boosting ICL scores.

**Sensitivity to Label Format** (Min et al., 2022b) also observes that the ICL performance relies on the label format. Replacing the label set with another random set will reduce the performance of ordinary auto-regressive LLMs. This can be explained in our theory that the model's output comes from a weighted voting of demonstrative sample labels $\{y_i\}$. If the label space is changed, the next token will also be misled to a different output space. So it is generally beneficial for ICL to ensure the demonstrative samples and test samples share an aligned label space and output format.

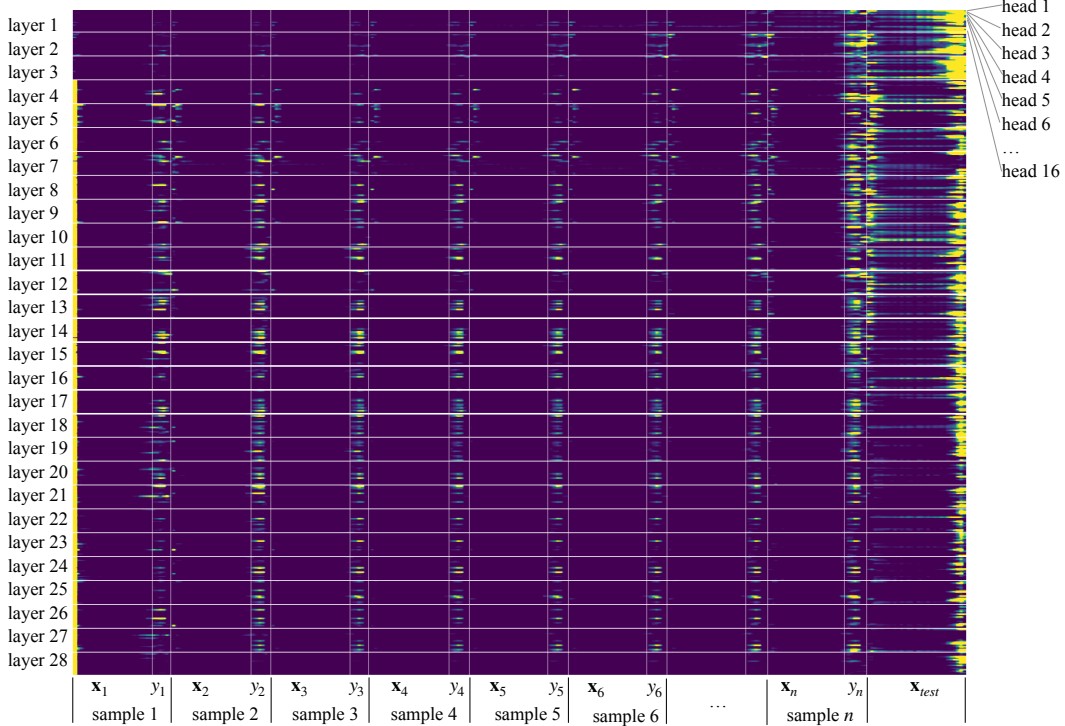

Figure 2: Averaged attention map over GLUE-sst2 test set. A portion of attention on demonstrative samples is generally focused on label positions $y_i$. This conforms to the intuition in Theorem 1 that the inference on in-context learning prompts is a weighted average over sample labels.

Table 1: Effect of OOD inputs on ICL on SST2 dataset.

| OOD-type | None | rare word | complex | typo |
|---|---|---|---|---|
| accuracy | 0.805 | 0.677 | 0.788 | 0.534 |

**Sample Selection** In Equation 6, the existence of the $\eta^2 \epsilon_\theta$ term implies that the demonstrations $[\mathbf{x}_i, y_i]$ should be sampled in a way close to $p_{\text{pre-train}}$. To verify this, we conduct an experiment by converting test inputs to semantically similar but rarer sentences. Specifically, on the SST2 dataset, we prompt GPT-3.5-turbo to generate semantically similar while rarer (i.e. OOD) expressions of inputs and use them to substitute the original dataset inputs. We consider three types of OOD types. "Rare word" is where words are substituted with rare synonyms. "Complex" is to express the original sentence in a more complex structure. "Typo" is where we require adding typos to the original input. The results are listed below. We see that compared with original scores ("None"), these OOD types more or less decrease the ICL accuracy, with "typo" having the largest effect, dropping the accuracy to near random.

**Bias from Pre-training Data** Theorem 1 implies that the final prediction depends both on in-context examples and prior knowledge from pre-training. This can be seen from the $\eta^2 \epsilon_\theta$ term in Equation 6, which comes from the pre-training information in Equation 16 in Appendix A. Specifically, this term describes how the pre-training distribution's bias affects the ICL prediction. This partially elucidates Kossen et al. (2023)'s finding that "ICL cannot overcome prediction preferences from pre-training."

**Remaining Challenges** However, we need to point out that there are still phenomena not explainable by our framework, as well as most previous explanations. One most mysterious one is the sensitivity to sample ordering (Lu et al., 2022) as a kernel regression should be order-ignorant, which no existing explanations (including (Xie et al., 2022; Akyürek et al., 2023)) take into account. Another intriguing question is that some work finds LLMs robust to perturbed or random labels(Kossen et al., 2023) while others find the opposite (Min et al., 2022b). We attribute such phenomena to the fact that LLMs also rely on a large portion of implicit reasoning in text generation and might benefit

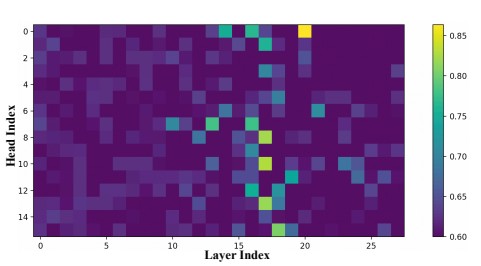 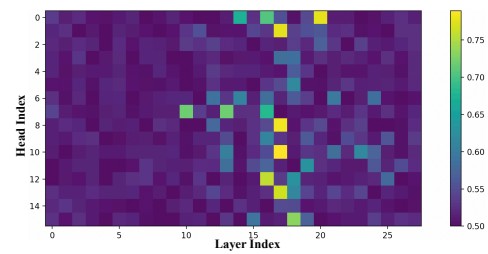

(a) Accuracy compared with model output $\hat{y}$.

(b) Accuracy compared with ground truth label $y_{\text{test}}$.

Figure 3: We use each head's attention weights on demonstrative samples to manually average sample labels $y_i$. These figures show that "reconstructed" outputs in some heads from layers 16∼21 matches LLM prediction with as high as 89.2% accuracy, and matches ground truth with 86.4% accuracy.

from linguistic cues in text prompts. Our theory provides a partial explanation, which needs to be combined with this implicit ability of LLMs to form a more comprehensive understanding of ICL.

## 5  EMPIRICAL ANALYSIS

In this section, we conduct empirical analysis on LLMs in order to verify our hypothesis. Because Equation 5 is only one special solution among infinitely many of its equivalent forms, and it also relies on the unknown HMM structure, it is infeasible to directly evaluate it on data. However, we can verify if it can predict observable behaviors on LLMs in experiments. Limited by computation resources in an academic lab, we analyze the GPT-J 6B model(Wang & Komatsuzaki, 2021) on one Tesla V100. It employs a decoder-only Transformer architecture. In this section, we use the validation set of the sst2 dataset as a case study, while results on more tasks can be found in Appendix B. We investigate the ICL behavior of LLMs from shallow to deep levels, and sequentially ask the following 4 questions: Do the attention heads collect label information $\mathbf{e}(y_i)$ as predicted? Does the attention-kernel analogy explain LLM's prediction? Can we actually explain the attention values as a kind of similarity? Can we find where algorithmic features $\mathbf{e}(y_i), T_{\mathbf{x}_i}$ are stored? The following sections answer these questions one by one.

### 5.1  WHERE ARE ATTENTIONS DISTRIBUTED DURING ICL?

First, we notice that Equation 2 implies that the LLM takes a weighted average over sample labels $y_i$ in ICL. Figure 2 shows how attention weights are distributed on in-context learning inputs $[S_n, \mathbf{x}_{\text{test}}]$. On each test point, we sample one ICL prompt and collect the attention map over previous tokens for predicting the next token. After getting the attention maps, as ICL samples $\mathbf{x}_i$ may have varied lengths, we re-scale the attentions on each $\mathbf{x}$ from $|\mathbf{x}|$ to a fixed 30-token length with linear interpolation. After aligning the attention lengths, we average all attention maps. The horizontal axis is the aligned positions on prompt $[S_n, \mathbf{x}_{\text{test}}]$. Each bar corresponds to one of 28 Transformer layers. Within each bar, each thin line is 1 out of 16 attention heads. Darker (blue) areas mean smaller averaged attention, while brighter areas indicate high attention.

In Figure 2, there are three major locations of attention masses. First, a majority of attention is focused on the final few tokens in $\mathbf{x}_{\text{test}}$, especially in the first 3 layers. This accords with previous observations that Transformer attentions tend to locate in a local window to construct local semantic feature for $\mathbf{x}_{\text{test}}$. Secondly, as also observed in previous studies, LLMs tend to allocate much attention on the first few tokens in a sequence to collect starter information. Finally and most intriguingly, we observe concentrated attention on each sample label tokens $\{y_i\}$. This phenomenon confirms an aggregation of label information in LLM ICL, in line with the prediction by Equation 5. Note that our explanation does not specify or limit the model from implementing kernel regression in a particular layer. In fact, an equivalent mechanism can occur in one or more layers as long as these (possibly redundant) results can be passed with skip connections to the final layer and aggregated.

## 5.2 CAN ATTENTIONS BE INTERPRETED AS KERNEL FUNCTIONS?

Now that we observe expected locations of attention weights on labels, we go on to verify if the LLM really predicts by averaging on labels as suggested by Theorem 1. We iterate over 16 heads and 28 layers, and insert their attention weights into Equation 5 to manually average the label distribution. This is similar in concept to a mind-reading experiment to predict one's next word using brain waves only (Wang & Ji, 2022). Specifically, for each attention head, we use the maximal attention value $a_i$ within the range of $[\mathbf{x}_i, y_i]$ as the kernel weight. Then on the ICL samples, we reconstruct the prediction as follows:

$$\tilde{y} = \arg\max \frac{\sum_{i=1}^{n} \mathbf{e}(y_i) a_i}{\sum_{i=1}^{n} a_i}$$

The resulting "reconstructed output" $\tilde{y}$ is compared for both LLM's actual prediction $\hat{y}$ and ground truth label $y_{\text{test}}$ to calculate its accuracy. Figure 3a and 3b plot the accuracy between $\tilde{y}$ and $\hat{y}$ and between $\tilde{y}$ and $y_{\text{test}}$ respectively. Interestingly, we spot the existence of several heads in layers 18~21 which demonstrate high accuracy in reconstruction. The highest of them (layer 17, head 10) achieves 89.2% accuracy on $\hat{y}$ and 86.4% accuracy on $y_{\text{test}}$. This result validates our hypothesis that some components in Transformer-based LLMs implement kernel regression. Note that this phenomenon happens within a few adjacent layers in the middle of the model. This is similar to our prediction in Section 4.1: not many attention layers are needed for kernel regression, as long as the required features have been computed by preceding layers. It is enough for the higher layers to only pass on the computed results.

## 5.3 WHICH SAMPLES RECEIVE HIGH ATTENTION?

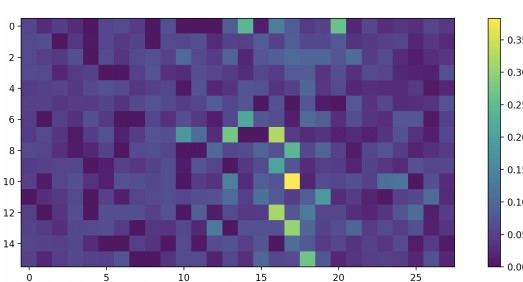

Figure 4: Pearson correlation between sample's attentions and *prediction similarity* $\text{sim}_{\text{pred}}(\mathbf{x}_{\text{test}}, \mathbf{x}_i)$ (Equation 8). $x$-axis are layers and $y$-axis are heads in each layer. Note the resemblence between this heatmap and Figure 3.

We go on and ask the question: if the LLMs use attention to implement kernel regression, *what kind of similarity does this kernel function evaluate?* From Equation 5, we see that the dot product is measuring similarity between $T_{\mathbf{x}}$, which encodes information necessary for HMM inference: $p(o|\mathbf{x}) = p_{\text{pre-train}}^{\top} T_{\mathbf{x}} B$. Therefore, we conjecture that the attention value $a_i$ between $\mathbf{x}_{\text{test}}$ and $\mathbf{x}_i$ correlates with their prediction similarity. Specifically, we define the *prediction similarity* as follows:

$$\text{sim}(\mathbf{x}_1, \mathbf{x}_2) = P(o|\mathbf{x}_1)^{\top} P(o|\mathbf{x}_2), \quad (8)$$

which is measured by applying LLMs on these texts *alone*, rather than in ICL prompt. Finally, we compute the Pearson correlation coefficient between $\text{sim}(\mathbf{x}_{\text{tes}}, \mathbf{x}_i)$ and each attention values on samples for each attention head. The results are shown in Figure 4. The absolute value of correlation is not high, as $P(o|\mathbf{x})$ is a dimension reduction to $T_{\mathbf{x}}$ and can lose and mix information. However, we can still note a striking similarity between it and Figure 3. This means that the heads responsible for ICL mostly attend to *prediction-similar* samples.

## 5.4 DO INTERMEDIATE FEATURES STORE INFORMATION USEFUL FOR KERNEL REGRESSION?

In this section, we go into a more detailed level, and investigate the question: *where do Transformer-based LMs store the algorithmic information needed by kernel regression?* To this end, we take out the intermediate *key* and *value* features in all layer heads and see if the correct information is stored in the correct locations. Note in Section 5.1, we observe that a major part of attention weights are located at the label position, so we focus on positions within $[-1, 3]$ relative to this position. Noticing the analogy we made at Section 4.1 that $k_i \sim \text{vec}(T_{\mathbf{x}_j})$ and $v_j \sim y_j$, we study two sub-questions: (1) whether *value* vectors encode label information $y_i$; and (2) whether *key* vectors encode LLM prediction information $P(o|\mathbf{x}_i)$. For each head, we conduct Ridge regression with $\lambda = 0.01$ to fit the task in these 2 questions. Results are presented in Figure 5. We can observe that, generally the high-attention position (y-axis = 0) indeed achieves the best accuracy. Figure 5b is intuitive, as tokens

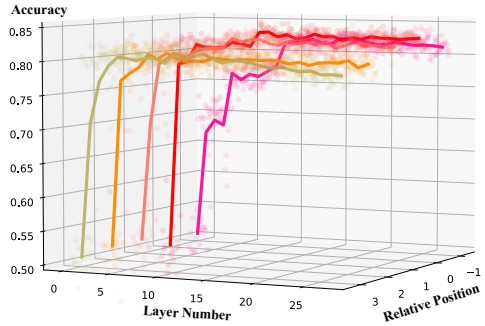
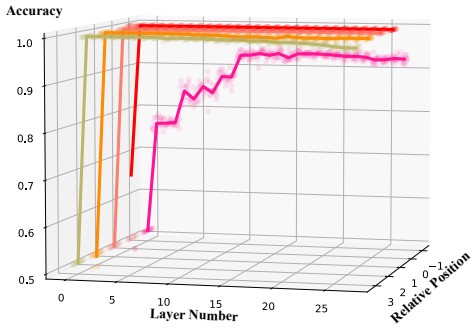

(a) Predicting $\arg\max_o P(o|\mathbf{x}_i)$ with *key* vectors.    (b) Predicting $y_i$ with *value* vectors.

Figure 5: Key and value vectors encode label and LLM prediction information at high-attention position. Here y-axis denotes the relative position to the high-attention position in each sample. Each sphere is an attention head. The curve shows average accuracy within each layer.

Table 2: Performance of explicit kernel regression (KR) and LLM ICL on downstream tasks.

| Method | sst2 | mnli | rotten-tomatoes | tweet_eval (hate) | tweet_eval (irony) | tweet_eval (offensive) |
|---|---|---|---|---|---|---|
| **GPT-J-6B ICL** | 0.805 | 0.383 | 0.671 | 0.539 | 0.519 | 0.542 |
| **ICL output reconstruction** | 0.797 | 0.597 | 0.683 | 0.713 | 0.722 | 0.753 |
| **Reconstructed performance** | 0.750 | 0.360 | 0.527 | 0.587 | 0.522 | 0.513 |
| **all-MiniLM-L6-v2** | 0.503 | 0.321 | 0.478 | 0.548 | 0.491 | 0.588 |
| **bert-base-nli-mean-tokens KR** | 0.523 | 0.325 | 0.502 | 0.545 | 0.479 | 0.597 |

at a position later than the label token $y_i$ can easily access the information of $y_i$ by self-attention. The slight drop at position +3 means that a longer distance introduces more noise to this information flow. Results in Figure 5a tell us that, although sentence $\mathbf{x}_i$'s starting position in ICL prompt is shifted and different from 0, $k_i$ is still strongly correlated with $P(o|\mathbf{x}_i)$, which indicates a sense of translation invariance. Overall, the results mean that, with the attention map distributed in Figure 2, LLM is able to use the attention mechanism to extract information regarding $T_{\mathbf{x}_i}$ and $y_i$ from *key* and *value* vectors effectively just as we described.

### 5.5 HOW WELL CAN OUR EXPLANATION RECONSTRUCT ICL PREDICTION?

Finally, we numerically evaluate our explanation on its ability to reconstruct the ICL predictions and tasks. We uniformly use 700 data in each validation set to represent tasks in a balanced way. We select the attention head that has the highest correlation with model predictions. This head is then evaluated on ICL prediction reconstruction and task performance on a held-out set of 300 data per task. In Table 2 we see that the ICL output reconstruction has an accuracy from 68% to 80% except for the harder task of MNLI. The task performance of reconstructed outputs matches the model's performance level. It also achieves similar or superior performance than kernel regression on sentence encoders such as all-MiniLM-L6-v2 and bert-base-nli-mean-tokens (Reimers & Gurevych, 2019).

## 6 CONCLUSIONS AND FUTURE WORK

In conclusion, our work provides a novel theoretical view to understand the intriguing in-context learning (ICL) capabilities of Transformer-based large language models (LLMs). We propose that LLMs ICL can be understood as kernel regression. Our empirical investigations into the in-context behaviors of LLMs reveal that the model's attention and hidden features during ICL are congruent with the behaviors of kernel regression. Furthermore, our theory also explains several observable phenomena in the field of ICL: why the retrieval of demonstrations similar to the test sample can enhance performance, the sensitivity of ICL to output formats, and the benefit from selecting in-distribution and representative samples. There are still remaining challenges in this topic, such as understanding the effect of sample orderings and the robustness to perturbed labels. These questions, along with understanding other perspectives of LLMs, are exciting questions for future research.

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

# A   PROOFS

*Proof.* First, we denote the kernel regression function

$$\hat{\mathbf{y}} = \frac{1}{n} \sum_{i=1}^{n} \left\langle \mathrm{vec}(T_{\mathbf{x}_{\mathrm{test}}}), \Sigma_{p_0}^{-1} \mathrm{vec}(T_{\mathbf{x}_i}) \right\rangle \mathbf{e}(y_i), \tag{9}$$

In expectation,

$$\mathbb{E}_{\mathbf{x}_i} \left\langle \mathrm{vec}(T_{\mathbf{x}_{\mathrm{test}}}), \Sigma_{p_{\theta^\star}}^{-1} \mathrm{vec}(T_{\mathbf{x}_i}) \right\rangle \mathrm{vec}(T_{\mathbf{x}_i})^\top = \mathrm{vec}(T_{\mathbf{x}_{\mathrm{test}}})^\top \Sigma_{p_{\theta^\star}}^{-1} \mathbb{E}_{\mathbf{x}_i} \mathrm{vec}(T_{\mathbf{x}_i}) \mathrm{vec}(T_{\mathbf{x}_i})^\top = \mathrm{vec}(T_{\mathbf{x}_{\mathrm{test}}})^\top. \tag{10}$$

As $\mathrm{vec}(\cdot)$ is a linear function, if we replace $\mathrm{vec}(T_{\mathbf{x}_i})^\top$ with $T_{\mathbf{x}_i}$:

$$\mathbb{E}_{\mathbf{x}_i} \left\langle \mathrm{vec}(T_{\mathbf{x}_{\mathrm{test}}}), \Sigma_{p_{\theta^\star}}^{-1} \mathrm{vec}(T_{\mathbf{x}_i}) \right\rangle T_{\mathbf{x}_i} = T_{\mathbf{x}_{\mathrm{test}}} \tag{11}$$

and then

$$\mathbb{E}_{\mathbf{x}_i} \left\langle \mathrm{vec}(T_{\mathbf{x}_{\mathrm{test}}}), \Sigma_{p_{\theta^\star}}^{-1} \mathrm{vec}(T_{\mathbf{x}_i}) \right\rangle P(y = Y | \mathbf{x}_i, p_{\theta^\star}) \tag{12}$$

$$= P(y = Y | \mathcal{S})^\top \mathbb{E}_{\mathbf{x}_i} \left\langle \mathrm{vec}(T_{\mathbf{x}_{\mathrm{test}}}), \Sigma_{p_{\theta^\star}}^{-1} \mathrm{vec}(T_{\mathbf{x}_i}) \right\rangle T_{\mathbf{x}_i} p_{\theta^\star} \tag{13}$$

$$= P(y = Y | \mathcal{S})^\top T_{\mathbf{x}_{\mathrm{test}}} p_{\theta^\star} \tag{14}$$

$$= P(y = Y | \mathbf{x}_{\mathrm{test}}, \theta^\star) \tag{15}$$

As $(\mathbf{x}_i, \mathbf{e}(y_i))$ can be seen as independent samples from $P(y = Y | \mathbf{x}_i, p_{\theta^\star})$, we can use Hoeffding's inequality and bound that, with $1 - \frac{\delta}{2}$ probability,

$$\| \hat{\mathbf{y}} - \mathbb{E}_{\mathbf{x}_i} \left\langle \mathrm{vec}(T_{\mathbf{x}_{\mathrm{test}}}), \Sigma_{p_{\theta^\star}}^{-1} \mathrm{vec}(T_{\mathbf{x}_i}) \right\rangle P(y = Y | \mathbf{x}_i, p_{\theta^\star}) \|_\infty \leq \sqrt{\frac{1}{2n} \ln \frac{4m}{\delta}}$$

Considering the difference between $\Sigma_{p_{\theta^\star}}$ and $\Sigma_{p_0}$, we see that

$$| \left\langle \mathrm{vec}(T_{\mathbf{x}_{\mathrm{test}}}), \Sigma_{p_{\theta^\star}}^{-1} \mathrm{vec}(T_{\mathbf{x}_i}) \right\rangle - \left\langle \mathrm{vec}(T_{\mathbf{x}_{\mathrm{test}}}), \Sigma_{p_0}^{-1} \mathrm{vec}(T_{\mathbf{x}_i}) \right\rangle | \tag{16}$$

$$= | \mathrm{vec}(T_{\mathbf{x}_{\mathrm{test}}})^\top (\Sigma_{p_{\theta^\star}}^{-1} - \Sigma_{p_0}^{-1}) \mathrm{vec}(T_{\mathbf{x}_i}) | \tag{17}$$

$$\leq \eta^2 \epsilon_\theta \tag{18}$$

Therefore,

$$\| \hat{\mathbf{y}} - P(y = Y | \mathbf{x}_{\mathrm{test}}, \theta^\star) \|_\infty \leq \sqrt{\frac{1}{2n} \ln \frac{4m}{\delta}} + \eta^2 \epsilon_\theta$$

Next we bridge $P(y = Y | \mathbf{x}_{\mathrm{test}}, \theta^\star)$ with $P(y = Y | [S_n, \mathbf{x}_{\mathrm{test}}], p_{\mathrm{pre\text{-}train}})$. Let $s_{\mathrm{test}}$ be the hidden state corresponding to first token of $\mathbf{x}_{\mathrm{test}}$, i.e., $\mathbf{x}_{\mathrm{test},0}$. We see that, the likelihood of $s_{\mathrm{test}} = s_{\theta^\star}$ is lower bounded by:

$$P(s_{\mathrm{test}} = s_{\theta^\star}, S_n | p_{\mathrm{pre\text{-}train}}) = \sum_{\theta \in \Theta} P(s_{\mathrm{test}} = s_{\theta^\star} | S_n, s_\theta) P(S_n | s_\theta) P(s_\theta | p_{\mathrm{pre\text{-}train}}) \tag{19}$$

$$= P(s_{\mathrm{test}} = s_{\theta^\star} | S_n, s_{\theta^\star}) P(S_n | s_{\theta^\star}) P(s_{\theta^\star} | p_{\mathrm{pre\text{-}train}}) \tag{20}$$

$$\geq P(S_n | s_{\theta^\star}) P(s_{\theta^\star} | p_{\mathrm{pre\text{-}train}}) \epsilon_r \tag{21}$$

$$(\text{Markov property}) \geq \left( \prod_{i=1}^{n} P([\mathbf{x}_i, y_i, o^{\mathrm{delim}}] | s_{\theta^\star}) P(s_{\theta^\star} | [\mathbf{x}_i, y_i, o^{\mathrm{delim}}], s_{\theta^\star}) \right) P(s_{\theta^\star} | p_{\mathrm{pre\text{-}train}}) \epsilon_r \tag{22}$$

$$(\text{by Assumption 1}) \geq \left( \prod_{i=1}^{n} P([\mathbf{x}_i, y_i] | s_{\theta^\star}) \epsilon_d \epsilon_r \right) P(s_{\theta^\star} | p_{\mathrm{pre\text{-}train}}) \epsilon_r \tag{23}$$

$$\geq \left( \prod_{i=1}^{n} P([\mathbf{x}_i, y_i] | s_{\theta^\star}) \right) P(s_{\theta^\star} | p_{\mathrm{pre\text{-}train}}) \epsilon_r^{n+1} \epsilon_d^n \tag{24}$$

For another task $\theta'$, $s_{\text{test}}$ is unlikely to be $s_{\theta'}$ because:

$$P(s_{\text{test}} = s_{\theta'}, S_n | p_{\text{pre-train}}) = \sum_{\theta \in \Theta} P(s_{\text{test}} = s_{\theta'} | S_n, s_\theta) P(S_n | s_\theta) P(s_\theta | p_{\text{pre-train}}) \tag{25}$$

$$= P(s_{\text{test}} = s_{\theta'} | S_n, s_{\theta'}) P(S_n | s_{\theta'}) P(s_{\theta'} | p_{\text{pre-train}}) \tag{26}$$

$$\text{(by Assumption 1)} \leq \left( \prod_{i=1}^{n} P([\mathbf{x}_i, y_i, o^{\text{delim}}] | \theta') \right) P(s_{\theta'} | p_{\text{pre-train}}) \tag{27}$$

$$\leq \left( \prod_{i=1}^{n} P([\mathbf{x}_i, y_i] | \theta') \right) P(s_{\theta'} | p_{\text{pre-train}}) \tag{28}$$

$$\tag{29}$$

Therefore, the Bayesian inference over $s_{\text{test}}$, is:

$$P(s_{\text{test}} = s_{\theta^\star} | [S_n, \mathbf{x}_{\text{test}}], p_{\text{pre-train}}) \tag{30}$$

$$= \frac{P(s_{\text{test}} = s_{\theta^\star}, S_n | p_{\text{pre-train}})}{P(S_n | p_{\text{pre-train}})} \tag{31}$$

$$= \frac{P(s_{\text{test}} = s_{\theta^\star}, S_n | p_{\text{pre-train}})}{\sum_\theta P(s_{\text{test}} = s_\theta, S_n | p_{\text{pre-train}})} \tag{32}$$

$$= \left( \sum_\theta \frac{\left( \prod_{i=1}^{n} P([\mathbf{x}_i, y_i] | \theta') \right) P(s_{\theta'} | p_{\text{pre-train}})}{P(s_{\theta^\star}, S_n | p_{\text{pre-train}})} \right)^{-1} \tag{33}$$

$$\geq \left( 1 + \min_{\theta \neq \theta^\star} \exp \left( \sum_{i=1}^{n} \ln \frac{P([\mathbf{x}_i, y_i] | \theta)}{P([\mathbf{x}_i, y_i] | \theta^\star)} + n \ln \frac{1}{\epsilon_d} + (n+1) \ln \frac{1}{\epsilon_r} + \ln \frac{1}{\|p_0\|_{-\infty}} \right) \right)^{-1} \tag{34}$$

$$\text{(with} 1 - \frac{\delta}{2} \text{prob.)} \geq \left( 1 + \min_{\theta \neq \theta^\star} \exp \left( -n\epsilon_{KL} + \sqrt{\frac{1}{n} \ln \frac{4}{\delta}} + n \ln \frac{1}{\epsilon_d} + (n+1) \ln \frac{1}{\epsilon_r} + \ln \frac{1}{\|p_0\|_{-\infty}} \right) \right)^{-1} \tag{35}$$

$$\geq 1 - \exp \left( -n\epsilon_{KL} + \sqrt{\frac{1}{n} \ln \frac{4}{\delta}} + n \ln \frac{1}{\epsilon_d} + (n+1) \ln \frac{1}{\epsilon_r} + \ln \frac{1}{\|p_0\|_{-\infty}} \right) \tag{36}$$

So that

$$\|\hat{\mathbf{y}} - P(y = Y | [S_n, \mathbf{x}_{\text{test}}], p_{\text{pre-train}})\|_\infty \tag{37}$$

$$\leq \sqrt{\frac{1}{n} \ln \frac{4m}{\delta}} + \eta^2 \epsilon_\theta + \frac{1}{\epsilon_r \|p_0\|_{-\infty}} \exp \left( -n(\epsilon_{KL} + \ln(\epsilon_d \epsilon_r)) + \sqrt{\frac{1}{n} \ln \frac{4}{\delta}} \right) \tag{38}$$

$$= O \left( \sqrt{\frac{1}{n} \ln \frac{4m}{\delta}} \right) + \eta^2 \epsilon_\theta \tag{39}$$

$$\square$$

# B RESULTS ON MORE TASKS

Besides the case study on SST2 dataset in Section 5, in this section we also provide experiment results on other tasks. In specific, we experiment on Rotten Tomatoes[1], Tweet Eval[2]'s (hate, irony and offensive subtasks) and MNLI[3]. The results are as follows.

## B.1 ROTTEN TOMATOES

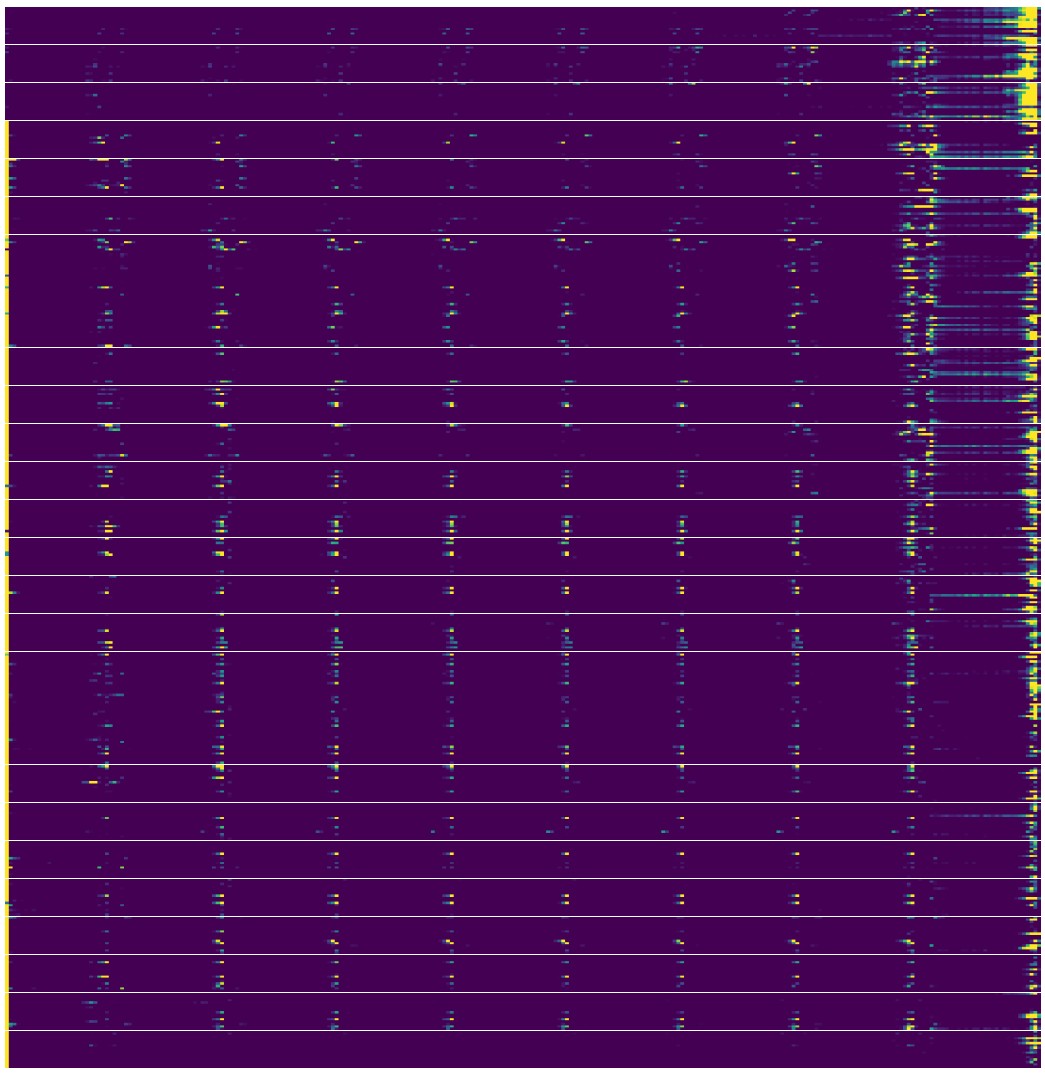

Figure 6: Averaged attention map over Rotten Tomatoes test set.

[1]https://huggingface.co/datasets/rotten_tomatoes/
[2]https://huggingface.co/datasets/tweet_eval/
[3]https://huggingface.co/datasets/glue/viewer/mnli_matched/test

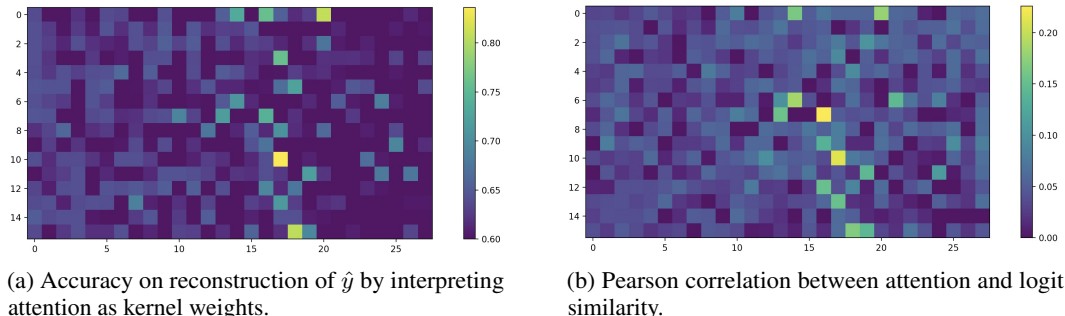

(a) Accuracy on reconstruction of $\hat{y}$ by interpreting attention as kernel weights.

(b) Pearson correlation between attention and logit similarity.

Figure 7: Interpreting attention values from kernerl regression perspective on Rotten Tomatoes dataset.

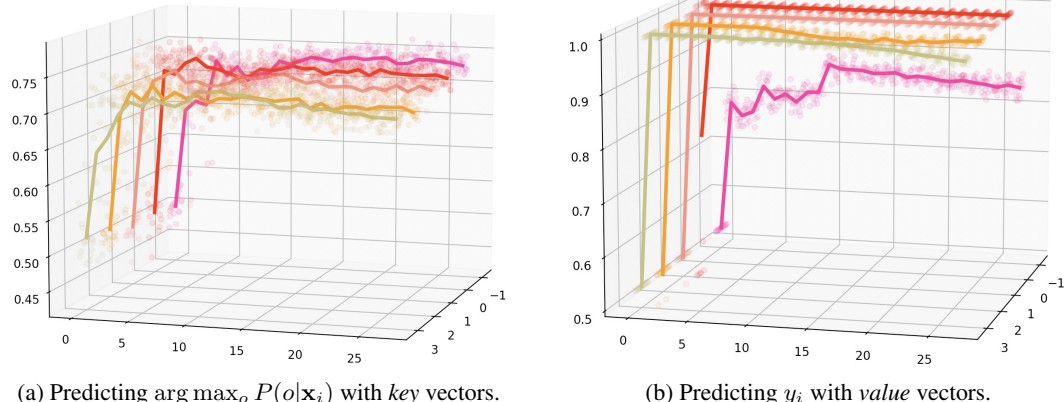

(a) Predicting $\arg\max_o P(o|\mathbf{x}_i)$ with *key* vectors.

(b) Predicting $y_i$ with *value* vectors.

Figure 8: Investigating information in key and value vectors on Rotten Tomatoes dataset.

## B.2 TWEET EVAL (HATE)

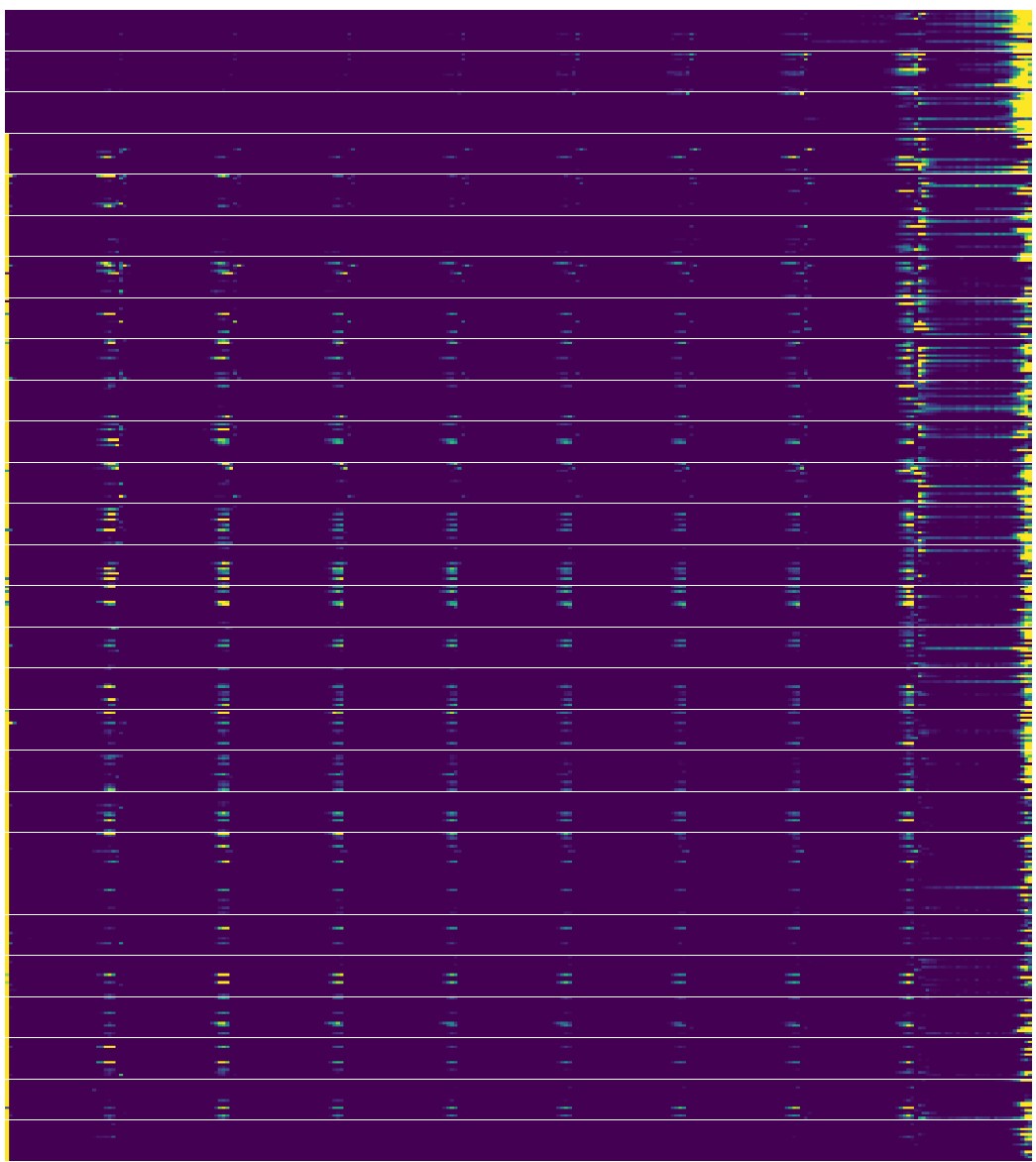

Figure 9: Averaged attention map over Tweet Eval (Hate) test set.

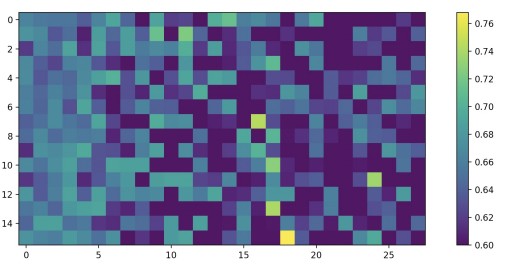 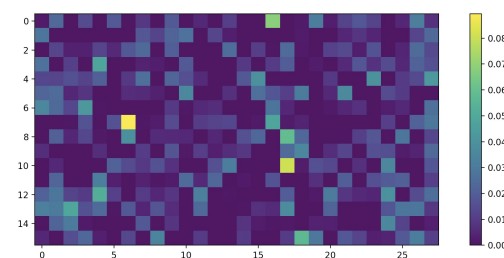

(a) Accuracy on reconstruction of $\hat{y}$ by interpreting attention as kernel weights.

(b) Pearson correlation between attention and logit similarity.

Figure 10: Interpreting attention values from kernerl regression perspective on Tweet Eval (Hate) dataset.

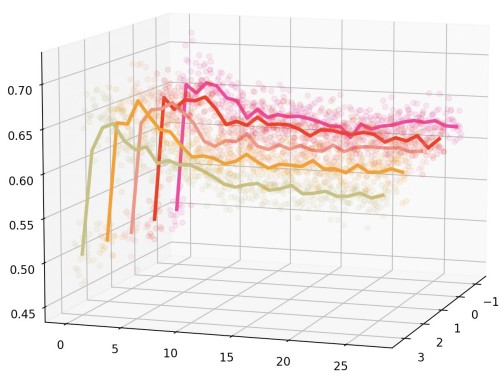 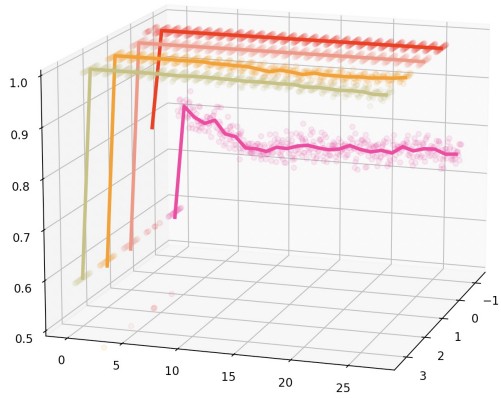

(a) Predicting $\arg\max_o P(o|\mathbf{x}_i)$ with *key* vectors.

(b) Predicting $y_i$ with *value* vectors.

Figure 11: Investigating information in key and value vectors on Tweet Eval (Hate) dataset.

## B.3 TWEET EVAL (IRONY)

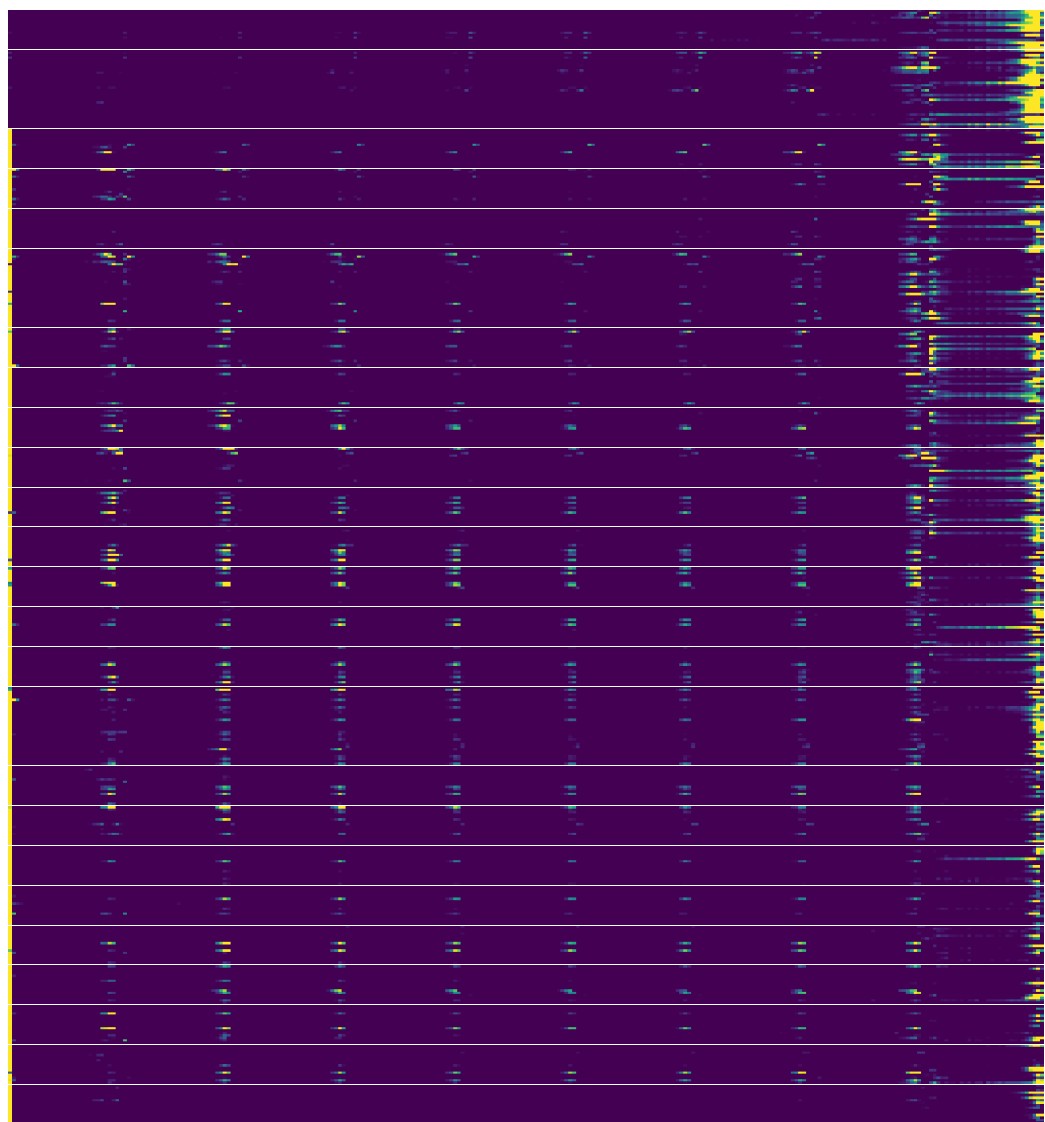

Figure 12: Averaged attention map over Tweet Eval (Irony) test set.

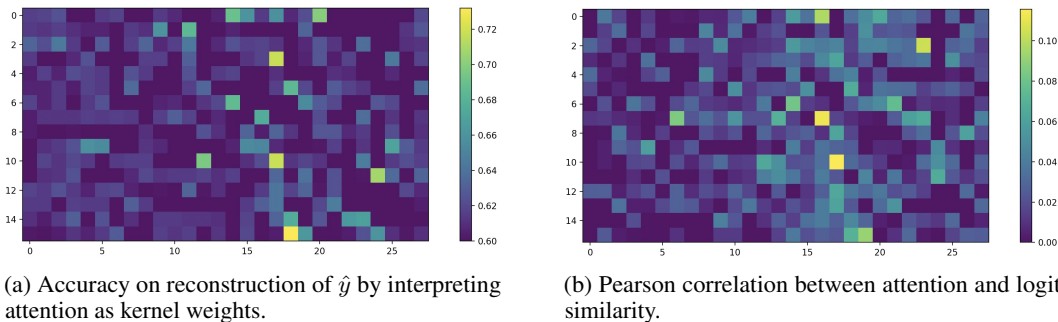

(a) Accuracy on reconstruction of $\hat{y}$ by interpreting attention as kernel weights.

(b) Pearson correlation between attention and logit similarity.

Figure 13: Interpreting attention values from kernerl regression perspective on Tweet Eval (Irony) dataset.

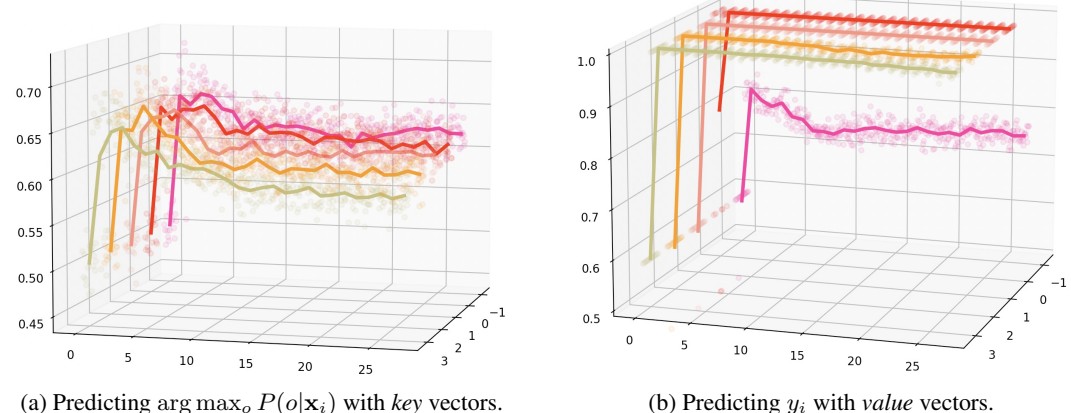

(a) Predicting $\arg\max_o P(o|\mathbf{x}_i)$ with *key* vectors.

(b) Predicting $y_i$ with *value* vectors.

Figure 14: Investigating information in key and value vectors on Tweet Eval (Irony) dataset.

### B.4 TWEET EVAL (OFFENSIVE)

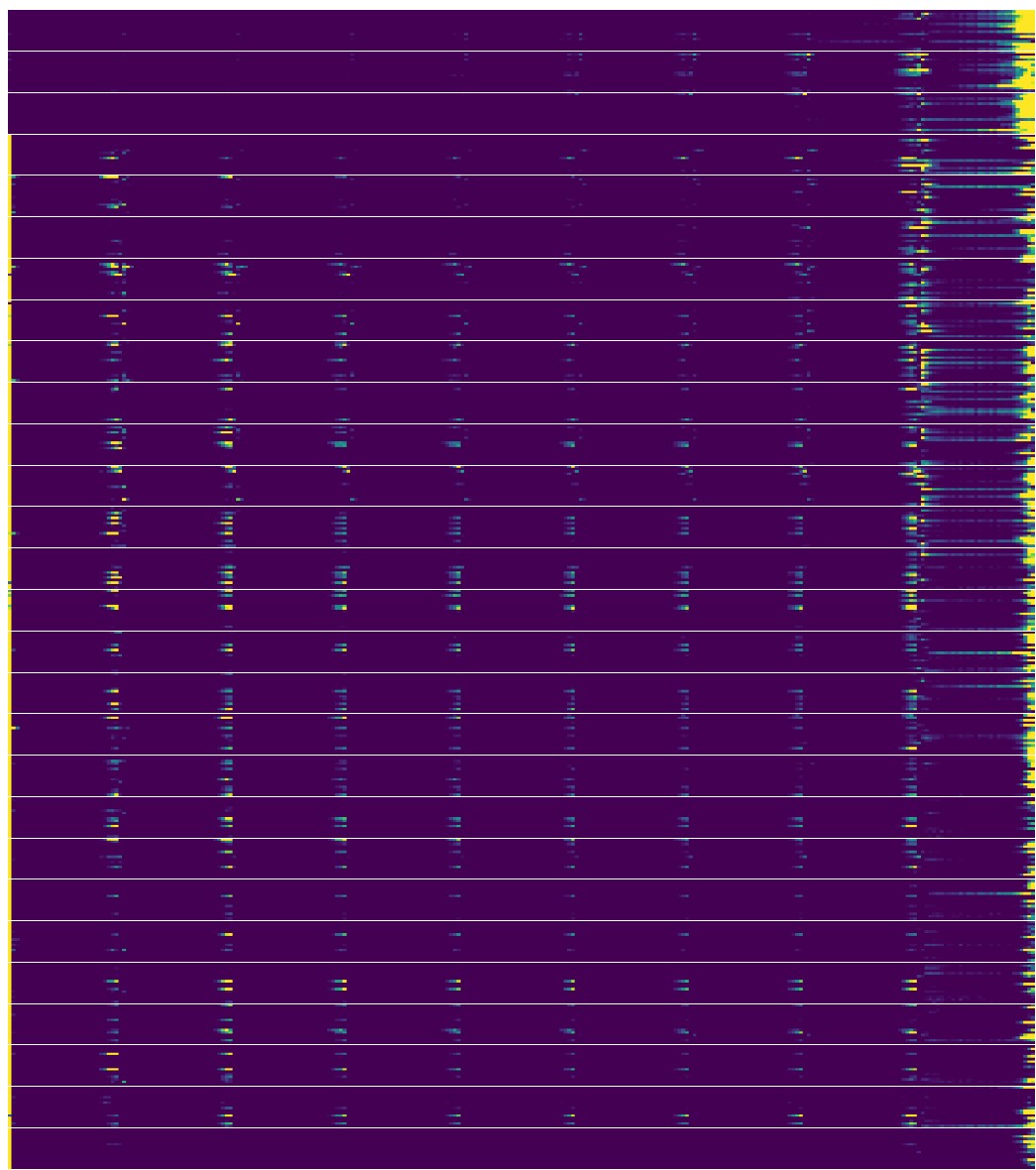

Figure 15: Averaged attention map over Tweet Eval (Offensive) test set.

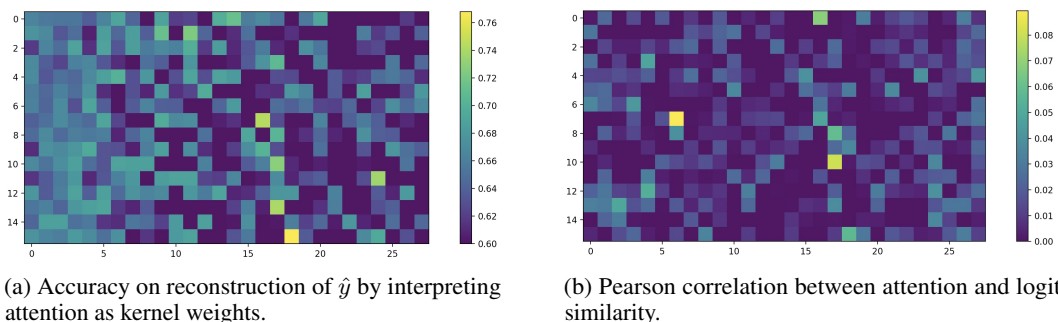

(a) Accuracy on reconstruction of $\hat{y}$ by interpreting attention as kernel weights.

(b) Pearson correlation between attention and logit similarity.

Figure 16: Interpreting attention values from kernerl regression perspective on Tweet Eval (Offensive) dataset.

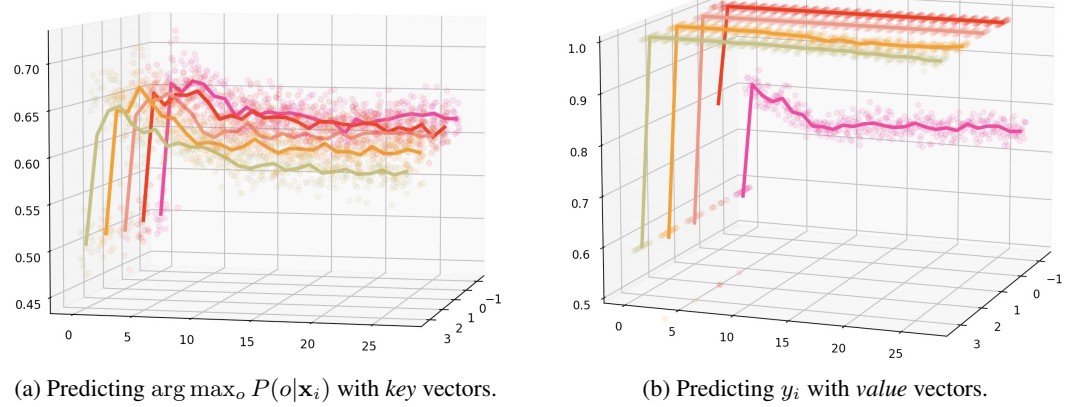

(a) Predicting $\arg\max_o P(o|\mathbf{x}_i)$ with *key* vectors.

(b) Predicting $y_i$ with *value* vectors.

Figure 17: Investigating information in key and value vectors on Tweet Eval (Offensive) dataset.

B.5 MNLI

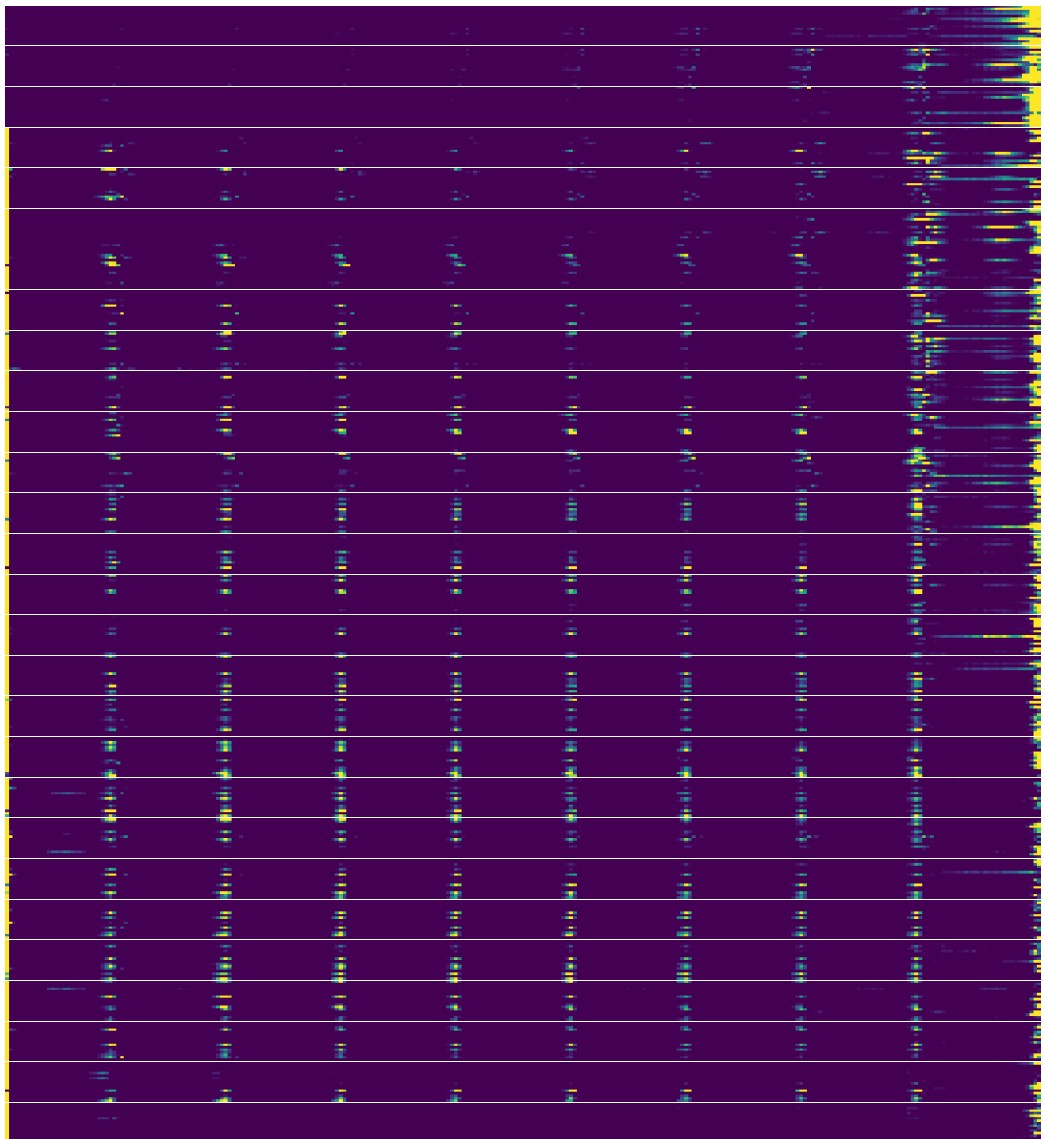

Figure 18: Averaged attention map over MNLI test set.

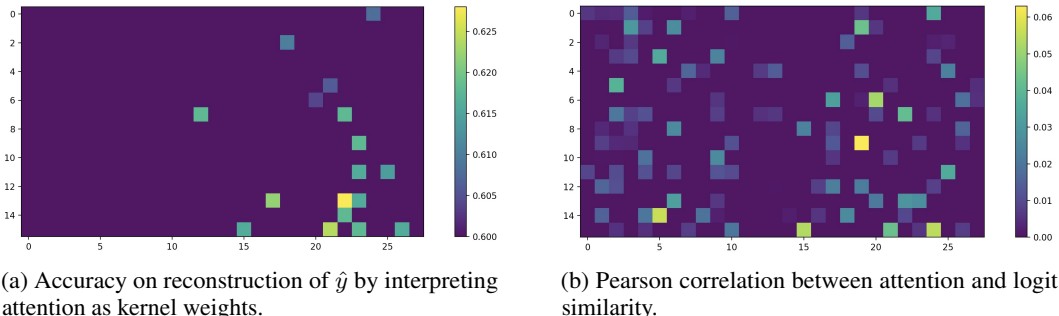

(a) Accuracy on reconstruction of $\hat{y}$ by interpreting attention as kernel weights.

(b) Pearson correlation between attention and logit similarity.

Figure 19: Interpreting attention values from kernerl regression perspective on MNLI dataset.

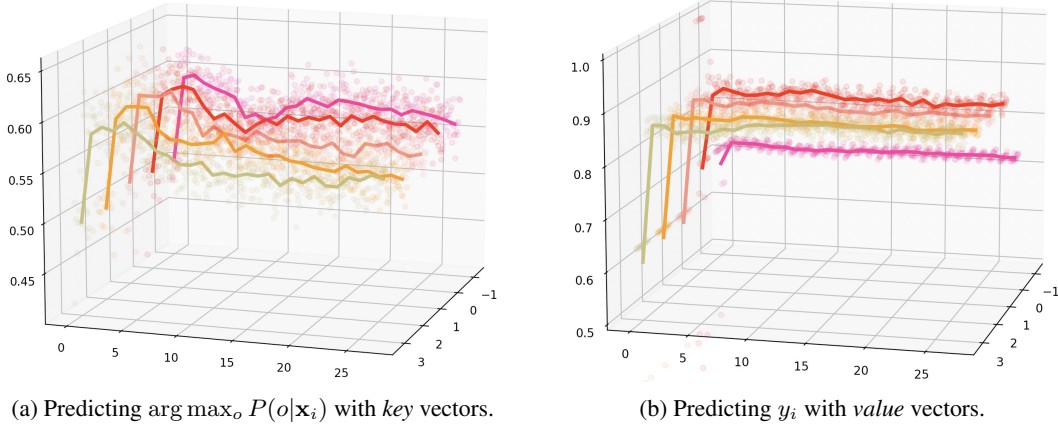

(a) Predicting $\arg \max_o P(o|\mathbf{x}_i)$ with *key* vectors.

(b) Predicting $y_i$ with *value* vectors.

Figure 20: Investigating information in key and value vectors on MNLI dataset.

## C  SYNTHETIC VERIFICATION

To verify the theoretical results, we conduct experiments on a synthetic HMM. We experimented on randomly parameterized synthetic HMMs with 8 tasks, 80 states, and 100 observations. We vary the number of samples and evaluate the proposed kernel regression on fitting the Bayesian posterior, which is the intuition in Theorem 1. Results are listed in the Table 3. We indeed see a decreasing loss and increasing accuracy with more demonstrative examples. The loss also converges to a non-zero value according to Equation 6.

Table 3: Accuracy and distance of predicting the Bayesian posterior on a synthetic HMM with Equation 5.

| #samples | 1 | 2 | 4 | 8 | 16 | 32 | 64 | 128 |
|---|---|---|---|---|---|---|---|---|
| distance | 1.322 | 0.942 | 0.519 | 0.218 | 0.163 | 0.085 | 0.093 | 0.083 |
| accuracy | 0.337 | 0.532 | 0.745 | 0.901 | 0.926 | 0.968 | 0.961 | 0.964 |

# D  RESULTS ON LLAMA-2

To demonstrate the generality of our conclusions across model architectures, we also conduct analysis on Llama-2 model (Touvron et al., 2023). The experiment setting is similar to Section 5.1 and 5.2. Visualizations are as follows. They generally show similar trends with Section 5.

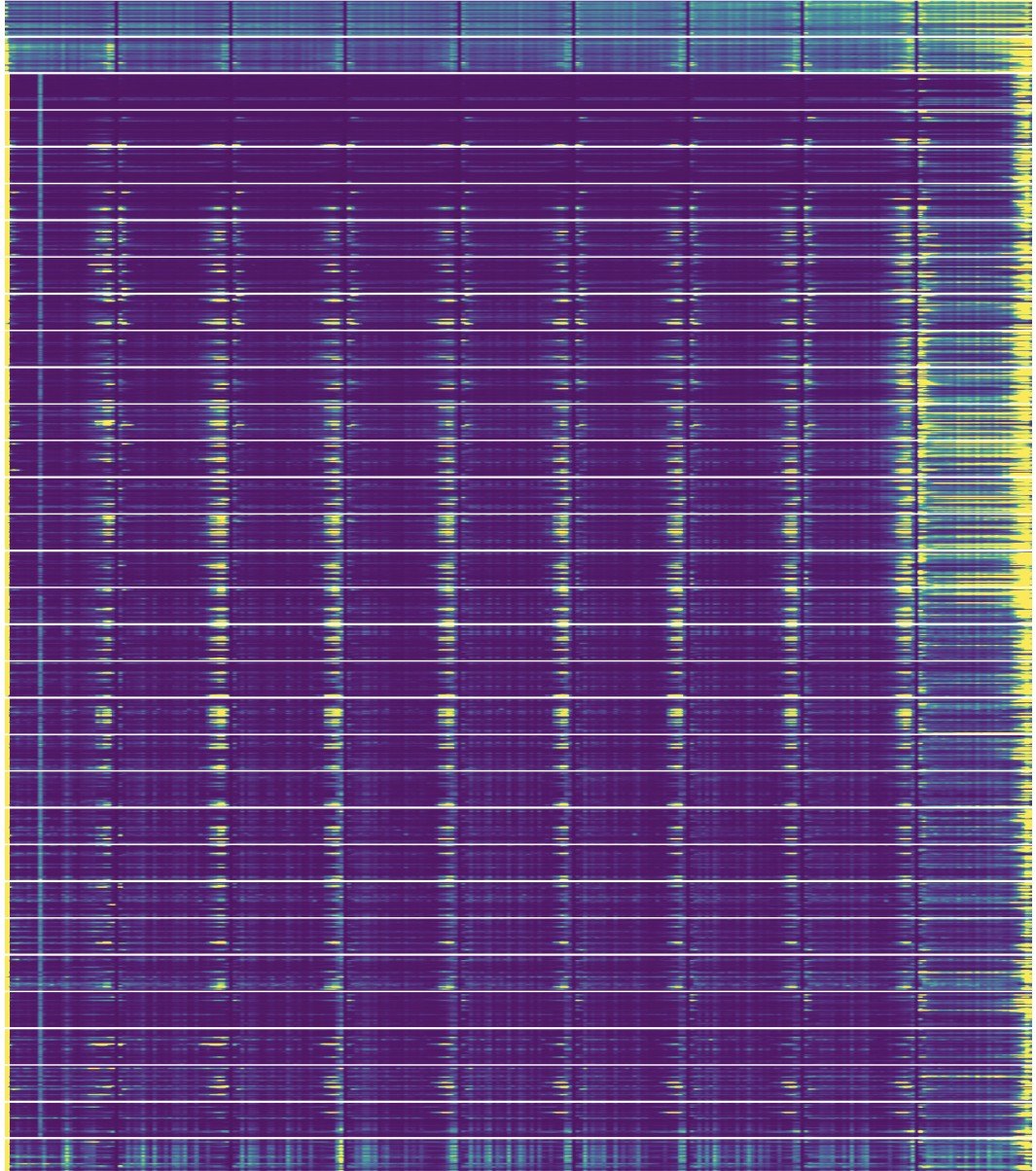

Figure 21: Averaged attention map over SST2 dataset on Llama-2 model.

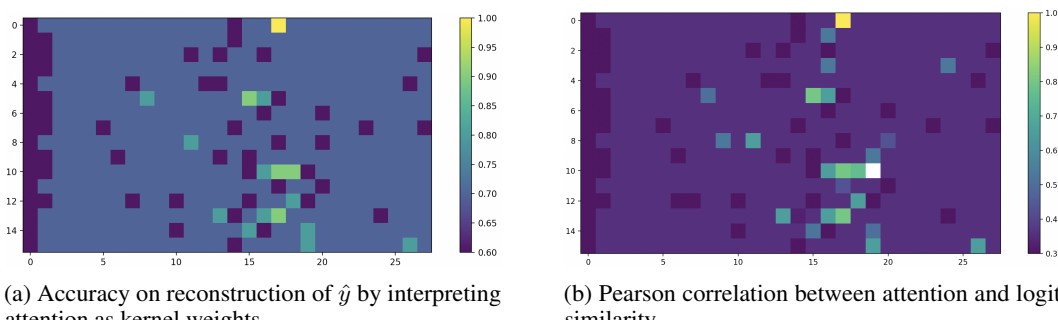

(a) Accuracy on reconstruction of $\hat{y}$ by interpreting attention as kernel weights.

(b) Pearson correlation between attention and logit similarity.

Figure 22: Interpreting attention values from kernerl regression perspective on Llama-2 model

# E    SINGLE DATUM VISUALIZATION

We also plot the visualization on a single datum as follows. It shows a similar but slightly sparser pattern to Section 5.1.

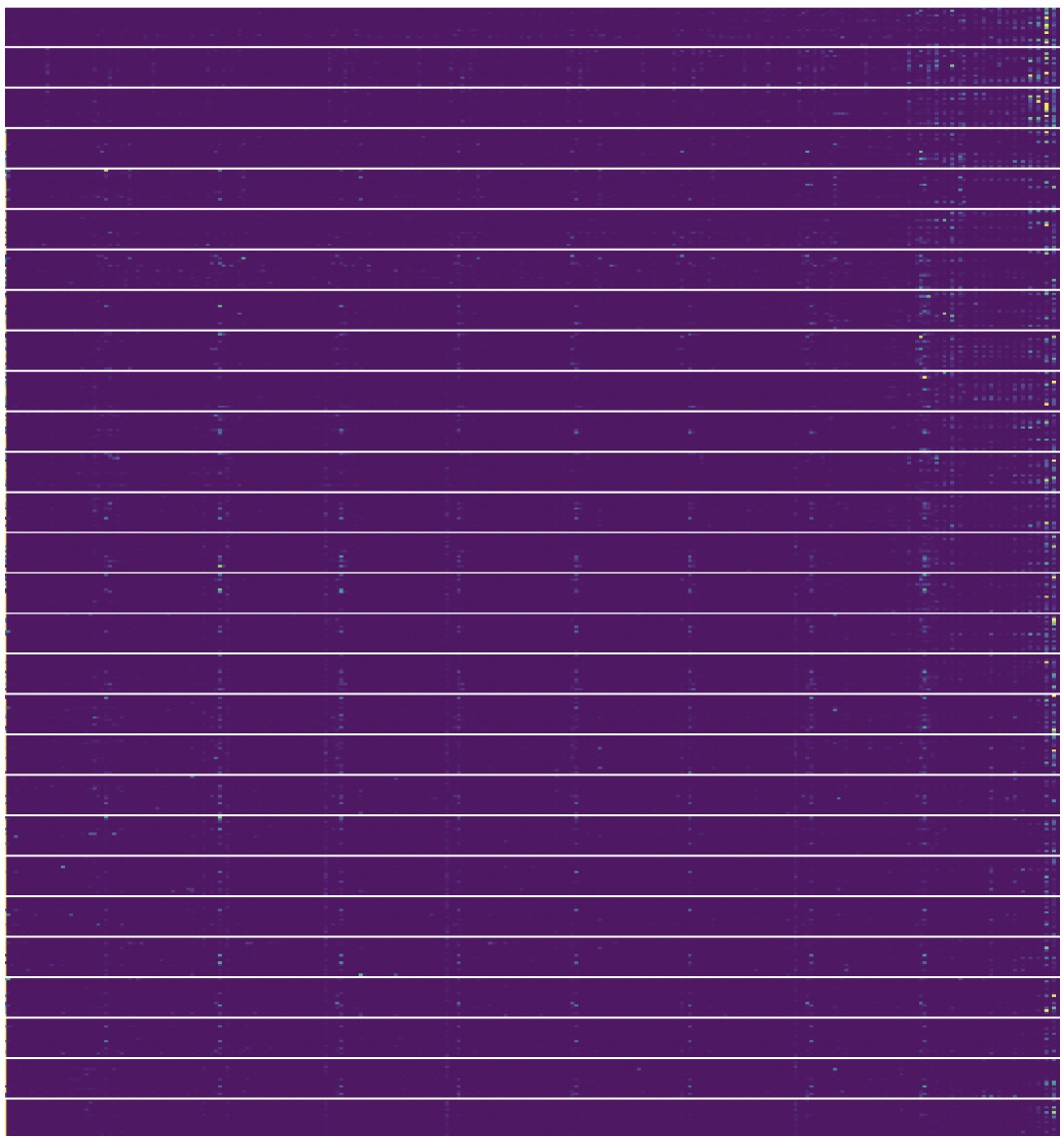

Figure 23: The attention map on a single datum.

