# OpenReview forum: "Explaining Emergent In-Context Learning as Kernel Regression"
_ICLR.cc/2024/Conference — Submitted to ICLR 2024_

### Official Review · Reviewer_iW9p · 2023-10-27

**Soundness:** 2 fair
**Presentation:** 3 good
**Contribution:** 3 good
**Rating:** 6
**Confidence:** 4

**Summary:**

The authors propose that In-Context Learning (ICL) in Large Language Models can be explained as kernel regression (KR).

Given a set of training examples, $\{(x_i, y_i)\}_{i=1}^N$ and a kernel that measures similarity between inputs, $K(x_i, x_j)$, KR makes predictions for a test input $x$ as a weighted average over the training points, where the weights come from the kernel similarity between test and training points, $\hat{y}=\frac{\sum_i y_i K(x_i, x)}{\sum_i K(x_i, x)}$.

Prior work proposes ICL implements Bayesian inference over latent concepts implied by the context. The authors show theoretically that, as the number of samples in the context increases, the Bayesian posterior converges to a KR.

The authors then appeal to intuitive arguments about the similarity of KR and the attention mechanism in Transformers.

To investigate whether transformers actually implement KR they claim KR allows one to understand why (a) retrieving similar examples improves ICL, (b) ICL is sensitive to the formatting of labels, (c) ICL is sensitive to the input distribution of examples.

Further, they provide a set of experiments that seeks to provide supporting evidence for KR in ICL.
They perform these experiments using a GPT-J-6B model on a variety of popular ICL tasks.

First, they display average attention maps from inputs to the test query across layers and heads of a transformer with a typical ICL task as input. They find attention values are concentrated around the labels of the input examples, which they take as evidence for a prediction similar to KR.

Further, they show instead of using the LLM to predict, one can also extract attention values and plug those into the KR equation as a substitute for the kernel. For attention values at particular depths/heads, this 'kernel regression with an LLM attention kernel' retains much of the accuracy of full ICL predictions. They take this as evidence that attention values measure similarity between inputs similar to a regression kernel.
They show that attention values between two inputs also correlate with the similarity of the labels predicted for them.

Lastly, they investigate if 'value' vectors of the attention mechanism encode label information and whether 'key' vectors encode information about which label is predicted.
(They take key and value vectors around the position of the input labels at ta variety of layers and heads.)
They find that a ridge regression model is able to predict both.

In summary, they claim 'the model’s attention and hidden features during ICL are congruent with the behaviors of kernel regression'.

**Strengths:**

Methods that provide useful interpretations of ICL are valuable to the community.
The authors interpretation of ICL as kernel regression (KR) is novel and their experiments are interesting.
I think that analogies between ICL and kernel regression are intuitively appealing, and potentially valuable to the community – if interpreted correctly.
The results suggest (clarifications pending) that some of the attention values in an LLM can be interpreted as a kernel similarity measure between ICL inputs.
The paper is written well and the presentation is mostly clear.

**Weaknesses:**

A) _It cannot be true that ICL directly implements KR._ I think the authors themselves are aware of this. For example, they note that the dependence of ICL to the order of examples is not congruent with this explanation.
However, instead of clearly rejecting that ICL directly implements KR, they just say this is 'mysterious' and point to the fact that prior work also ignores this.
(For the same reason, ICL also does not implement exact Bayesian inference.)

Now, the authors do not directly claim anywhere that "ICL=KR" holds exactly. However, I believe their language, and that of prior work for that matter, is highly misleading.
For example, they write that "LLMs can simulate kernel regression" or that "the model’s attention and hidden features during ICL are congruent with the behaviors of kernel regression".
This language is not wrong, however, I find it vague and intentionaly misleading.
Already, I hear too many people claim "ICL=Bayesian Inference" or "ICL=Gradient Descent" because papers are written with similar misleading language.

I do think there are clear benefits to analogies between ICL and KR.
However, I implore the authors to make their phrasing more honest.
The field does not benefit when careless readers of the paper take away that "LLMs can simulate kernel regression".

Instead, I would like to see a paper that explores analogies between ICL and KR: what can we learn from them and where do they fall short?

B) I think S. 4.2 'Insights provided by the explanation' requires significant improvement.

B1) Another reason why ICL does not implement KR directly are the label flipping experiments of Wei et al. (2023, https://arxiv.org/abs/2303.03846) and Kossen et al. (2023, https://arxiv.org/abs/2307.12375).
They show that LLM performance generally suffers when 'flipping' the label relationship, e.g. changing labels from positive/negative to negative/positive for a sentiment analysis task.
However, the kernel regression equation would predict that the performance of ICL does not depend on what tokens are used to encode the labels.
I would appreciate it if the authors would add these observations to their discussion section.


B2) Relatedly, I cannot follow the authors when they say that kernel regression predicts that 'ICL performance relies on the label format'.  I would appreciate if you could specify which of the experiments of Min et al. (2022b) you are referring to here. If it is their experiments in S. 5.2 which they call 'removing the format' (Impact of Input-Label Pairing) I would suggest you rephrase the paragraph here. In their experiments,  Min et al. here completely remove either the labels or demonstrations from the input. Such input is entirely incompatible with KR.

B3) I agree that KR provides an explanation for why similar samples help and why OOD samples (with high distance input space to the test query) may be harmful. However, I think it is highly misleading to claim that Min et al. (2022b) show that 'out-of-distribution (OOD) demonstrative samples will degrade ICL accuracy'.  It is important to note that what they call 'OOD' does not conform to standard expectations. In their OOD experiments they construct a set of input features that are 'randomly sampled from an external corpus', i.e. they completely destroy the input label relationship. This is different for classical OOD, e.g. samples from the same p(X, Y) but for rare or unusual X.

B4) Lastly, in their 'Remaining Challenges' section, the authors refer to Min et al. (2022b)'s claim of 'LLM’s robustness to perturbed or random labels'. Luckily for the authors, it seems these claims do not always hold up to scrutiny (which is good for analogies between ICL and KR), see Kossen et al. (2023, https://arxiv.org/abs/2307.12375).

B5) The MNLI ICL performance numbers you report in Table 2 are not better than random guessing performance (\~30%). Yet one of the attention head KR variants obtains performance _much_ better than random guessing (\~97%). If ICL performance is not good overall, I would not expect it to be outperformed by the 'head KR' variant. This makes me suspicious that there is some sort of overfitting going on here. I.e. the space of attention values is large (num_layers x num_heads) and one of them just happens to predict the true labels well. Do you have train/test splits for selecting and evaluating the KR head variant? Do you evaluate this over the full datasets or just a subset?

C) It would be great to see experiments with additional models, e.g. LLaMA/Falcon/Mistral models, to show your experimental observations do not depend on the GPT-j architecture.

D) The abstract/title/intro are somewhat misleading. The title focuses on 'emergent' ICL and the abstract asks 'how pretrained LLMs acquire such capabilities' and later the authors claim they 'make a preliminary step towards understanding ICL as a special case of LM capacity emergence'.
I do not think this work provides any understanding towards _how_ ICL emerges.
Instead they propose a mechanism for how ICL functions _given_ that is has emerged already.
I would suggest the authors remove references to such claims.


D) I am not too familiar with proofs around KR and Bayesian inference. However, could you elaborate where Theorem 1 depends on examples being presented 'in-context'? It seems to me that, perhaps, theorem 1 could hold true for all inference problems?


E) The connection between kernel regression and Transformer attention is currently quite hand-wavy, with one paragraph pointing at similarities in the equation. Perhaps the biggest difference is the change from a kernel to the $\exp(q, k)$ from the attention. The authors say this "can be regarded as a kernel trick". I think that more discussion here would strengthen the submission. How do the differences here change prediction behavior? Does softmax attention imply a particular form for the kernel?


F) what is $T_x$ in Equation 4? I don't see a definition for it in the main text. Later you write 'semantic vectors of sample inputs', but that does not help. Is it related to the transition matrix?

G1) Figure 2: We observe that attention is high around the labels of the examples, as well as the input features of the test query.  You write that 'This conforms to the intuition in kernel regression explanation in Theorem 1 that the inference on in-context learning prompts is a weighted average over sample labels.'
However, looking at kernel regression, I would have expected the attention to compare _different input features_, i.e. the kernel is $K(x, x_i)$ and not $K(x, y)$.
I would appreciate if you could comment on this.

Further, there seems to be a disconnect between the simplicity of kernel regression equation (features are directly available and separate from labels) and the reality of ICL (inputs are just a sequence of tokens, with features following labels, features (and labels) consist of multiple tokens).
I would like to see the authors engage with these differences.

For example, to implement KR with a Transformer, it seems to me one would need three steps:
1) compute similarity between example features and query features
2) pick out labels of the inputs
3) then weight labels by similarity

I think the current discussion of the authors on how ICL could implement KR is insufficient.

G2) Here is an experiment I would like to see.  Can the KR head approximation introduced in S. 5.2 still recover the models accuracies if you randomly permute the input features? I.e. for each input `[(x1, y1), (x2, y2), (x3, y3), ...] + [x_{test}]` and you permute the features `[(x3, y1), (x10, y2), (x5, y3) ,.... ] + [x_test]`.
Obviously, ICL performance will suffer from this. However, if the KR hypothesis holds true, you should still be able to approximate the models predictions from attention values just as well, as this just corresponds to a permutation of the kernel terms. However, can you actually reproduce Figure 3 with this setup?
(This would also be interesting to explore for the label _flipping_ experiment I mentioned above.)

H) It would be interesting to see Figure 2 for individual examples.

I) "This is similar to our prediction in Section 4.1: not many attention layers are needed for kernel regression, as long as the required features have been computed by preceding layers" --> That makes sense. However, it how many layers you need to compute adequate features could vary a lot between tasks. Therefore, the KR hypothesis does not allow you to make any hypothesis about the number of layers needed for ICL. Maybe if you had a toy experiment where the embeddings were perfect already?

J) "whether key vectors encode LLM prediction information P(o|xi)." --> Based on Equation 5, I would have expected the key vectors to encode information about the input features x_i and not the predicted labels.



## Nits

K) "They focus on a different goal and setting and propose to substitute attention mechanisms for more calibrated predictions" → I've looked at the related work here, and I agree it is quite different. However your description here is rather vague. I suggest you replace this with something mroe concrete.

L) Equation 7: Should the sum here be over $j$?

M) Please position the captions for tables above the table as per style guide.

N) Please GPT-j instead of just providing a link. The authors prefer the citation given at https://github.com/kingoflolz/mesh-transformer-jax.

O) Please at labels to the axes of Figure 3 and 5.

**Questions:**

P) "Specifically, for each attention head, we use the maximal attention value ai within the range of [xi, yi] as the kernel weight" -->  Interesting, based on your previous motivation I would have thought you would just take the attention value at y_i? why maximal value? If this is an ad-hoc choice, make that clear. How well does average work?

[edit] Changed score to 6 after author rebuttal. (It seems the authors cannot see my reply to their rebuttal. My score increase is conditioned on the promise of the authors to fully clear up my concern A in the camera ready version. I hope they will be able to see my full response by then.)

---

> ### Author Response · Authors · 2023-11-23
> **Author Response (1/2)**
>
> First, we thank the reviewer very much for this detailed and high-quality review. We are glad that the reviewer finds the interpretation “intuitively appealing” and experiments “interesting”. We respond to the comments as follows:
>
>
> **A**
>
> We agree with the reviewer that adopting a more rigorous language helps better position the paper. Inspired by your comments, we revised the presentation of the paper in the abstract and introduction. We propose to state that the work is “Understanding In-Context Learning from A Kernel Regression Perspective”, which avoids the problems in “explanation” but also takes our computational feature analysis into consideration. In the revision, we updated the abstract and introduction. As ICLR now forbids title updates during rebuttal, we also plan to revise the title at the camera-ready phase.
>
> **B1**
>
> Thanks for pointing to this interesting phenomenon under label flipping. We see it as a great discussion to improve clarity. In the revised Section 4.2, we explain that Theorem 1 implies that the final prediction depends both on (1) in-context examples and (2) prior knowledge from pre-training. This can be seen from the $\eta^2 \epsilon_\theta$ term in Equation (6), which comes from the pre-training information in Equation (16-18) in Appendix A. Specifically, this term describes that the bias given by the pre-training distribution affects the ICL prediction, even if labels are flipped. This gives a partial elucidation of why ICL still depends on what tokens are because they might conflict with the bias in pre-training corpora. This will also align with Kossen et al.’s finding that “ICL cannot overcome prediction preferences from pre-training.”
>
> **B2**
>
> The referred-to part is Section 5.2, “Impact of the label space” in Min et al., where they substitute labels with a random set of English words. As responded in (B1), using a label space different from the pre-training distribution will make the KR term diverge from or even conflict with the pre-training bias, thus influencing the final performance.
>
> **B3**
>
> Thanks for suggesting a correction, and we agree that a better experiment on evaluating the effect of ODD examples is beneficial. We conduct an experiment by converting test inputs to semantically similar but rarer sentences. Specifically, on the SST2 dataset, we prompt GPT-3.5-turbo to generate semantically similar while rarer (i.e. OOD) expressions of inputs and use them to substitute the original dataset inputs. We consider 3 types of OOD types. “Rare word” is where words are substituted with rare synonyms. “Complex” is to express the original sentence in a more complex structure. “Typo” is where we require adding typos to the original input. The results are listed below. We see that compared with original inputs (“None” category), these OOD types more or less decrease the ICL accuracy, with “typo” having the largest effect, dropping the accuracy to near random. This accords with our insight in Section 4.2 that OOD samples can negatively affect ICL performance. We add these results in the revision.
>
> | OOD-type | None | rare word | complex | typo |
> | --- | --- | --- | --- | --- |
> | accuracy | 0.805 | 0.677 | 0.788 | 0.534 |
>
>
> **B4**
>
> We thank the reviewer for referring to this valuable evidence and will add this to the discussion.
>
> **B5**
>
> The reviewer raises an interesting concern about the performance of the MNLI dataset. We follow the reviewer’s advice to eliminate the overfitting problem. As some MNLI does not provide test labels, and some tasks only have ~1000 pieces of data, we uniformly use 700 data in each validation set to represent tasks in a balanced way. We select the attention head that has the highest correlation with model predictions. It is then evaluated on model prediction reconstruction and task performance on a held-out set of 300 data per task. We see that the single-head reconstruction has an accuracy from 68\% to 80\% except on MNLI which is harder. The task performance of reconstructed outputs matches the model’s performance level. We revised the paper and merged Tables 1 and 2 accordingly.
>
> | metric | sst2 | mnli | rotten-tomatoes | tweet_eval (hate) | tweet_eval (irony) | tweet_eval (offensive) |
> | ----------- | ----------- | ----------- | ----------- | ----------- | ----------- | ----------- |
> | prediction reconstruction | 0.797 | 0.597 | 0.683 | 0.713 | 0.722 | 0.753 |
> | task performance | 0.750 | 0.360 | 0.527 | 0.587 | 0.522 | 0.513 |
>
>
> **C**
>
> We went on to experiment on the LLaMa model. Results generally show similar trends and are added to Appendix D.
>
> **D1**
>
> We agree and plan to modify the statements. Please see responses to your comment (A) for more details.

---

> > ### Author Response · Authors · 2023-11-23
> > **Author Response (2/2)**
> >
> > **D2**
> >
> > The connection between Theorem 1 and “in-context” examples is Equation (19-24), which describes the effect of “in-context” examples $[\mathbf{x}_i, y_i]$ on the conditional probability of the test input. This is a unique challenge in LLM ICL, where demonstrative examples are combined in the same sequence as the testing input. In other words, LLMs are fed with the whole sequence in a uniform manner, but we provide little information on how the ICL sequence is constructed.
> >
> > **E**
> >
> > The reviewer points to a valuable place for clarification. As $e^{q^\top k}$ (i.e., the Gaussian kernel or exponential kernel) is positive definite, it satisfies Mercer’s condition (https://en.wikipedia.org/wiki/Mercer's_theorem#Mercer's_condition), and Mercer’s theorem guarantees the existence of a kernel function. In case the kernel might be of infinite dimension (as is also the case for many other fields), a feature function that approximates the real kernel reasonably well is usually regarded as a “kernel trick”. We revised and added this part to the discussion.
> >
> > **F**
> >
> > $T_{\mathbf{x}} = \prod_{i=0}^{l-1} \text{diag}(\mathbf{p}_{x_i})T $ is defined in the paragraph under Equation (3). It is indeed related to the transition matrix and can be used to compute the conditional probability after a sequence of observations $\mathbf{x}$.
> >
> > **G1**
> >
> > This is another interesting question. First, your intuition is right in that the attention “compares different input features”. This does not conflict with the current attention pattern because the label position contains information from the whole example $[\mathbf{x}_i, y_i]$ instead of only information $y_i$, as investigated in Section 5.4. Second, the attention weights need to be centered at label positions because it is the only place that $v$ access the label information $y_i$ because of the causal mask.
> >
> > About the second question on how representations of tokens $\mathbf{x}_i$ can be computed in initial layers, this question requires describing the mathematical structure of linguistic data, which is an open challenge due to the lack of a suitable math framework. This is also the reason why many previous papers, such as Akyürek et al. (https://arxiv.org/abs/2211.15661), only study length-1 inputs.
> >
> > **G2**
> >
> > The reviewer raises an intriguing thought experiment. Unfortunately, we find that LLMs produce nonsense generations no matter if we permute a certain layer of multiple consecutive layers. We attribute this to LLMs’ sensitivity to intermediate feature distributions and intolerance to feature manipulations.
> >
> > **H**
> >
> > We plot the first example in the dataset for Figure 2 in Appendix E. It shows a similar but slightly sparser pattern to Section 5.1.
> >
> > **I**
> >
> > We agree with the reviewer in this regard, in the sense that the theory is only an existence statement but not constructive, so in practice, how many layers of features are needed should depend on the specific data/task in hand.
> >
> > **J**
> >
> > The key vectors indeed encode the input features for $\mathbf{x}_i$. Additionally, in LLMs, which are pre-trained to predict the next tokens, a useful feature is encouraged to correlate with the next token predictions by the learning objective.
> >
> > **K-O**
> >
> > Thank you for pointing out the typos, and we revise the submission accordingly.
> >
> > **P**
> >
> > The reviewer raises another interesting question. This technical choice of using max attention was from the intuitive observation in Figure 2, that only a few high-value attention values exist in the attention map. We also tried using mean or sum in place of the maximum attention and found it to have little effect on Figures 3 and 4. This is probably due to the fact that the majority of attention weights are concentrated in the few high-value positions in each sample.
> >
> >
> > Finally, we hope that this addresses your concerns. If you find our response helpful in answering your questions, we kindly request you to consider updating the reviews.

---

### Official Review · Reviewer_oa7R · 2023-10-29

**Soundness:** 3 good
**Presentation:** 3 good
**Contribution:** 3 good
**Rating:** 6
**Confidence:** 3

**Summary:**

The paper investigates the in-context learning (ICL) capability of large language models (LLMs) and aims to understand how LLMs acquire this capability after pre-training on a general language corpus. The authors propose a hypothesis that LLMs can simulate kernel regression with internal representations when faced with in-context examples. They prove that Bayesian inference on in-context prompts can be asymptotically understood as kernel regression. Through empirical investigation, they find that the attention and hidden features in LLMs during ICL exhibit behaviors similar to kernel regression. The paper's theory provides insights into various phenomena observed in the ICL field, including the benefits of retrieving similar demonstrative samples, sensitivity to output formats, and the advantages of selecting in-distribution and representative samples for ICL accuracy.

**Strengths:**

The paper addresses a timely and important question regarding the acquisition of in-context learning capability in large language models (LLMs). By adopting a kernel regression perspective, the authors provide an insightful explanation for empirical phenomena observed in the literature, such as the sensitivity to label format and the poor performance caused by out-of-distribution (OOD) demonstrative examples. This theoretical framework sheds light on the underlying mechanisms of in-context learning in LLMs and offers valuable insights into the field.

**Weaknesses:**

As mentioned in the paper's limitations, the current Kernel Regression hypothesis falls short in explaining the impact of sample orderings and the robustness to perturbed labels. Additionally, there is a gap between the simulated experiments and real sequential data and models in the current experiments. However, given the immense challenge of learning the in-context learning (ICL) capability of large language models (LLMs), I appreciate the intuitive nature of the Kernel Regression hypothesis presented in this paper. It aligns with the idea that LLMs infer results based on information from similar demonstrations (represented by $x_i$) and compute a weighted sum of their labels ($y_i$).

Minor issues:

1. The symbol $p(o)$ appears to have an incorrect LaTeX expression under Equation 3 in this paper.
2. The penultimate sentence in the discussion on Sensitivity to Label Format may contain grammatical errors.

**Questions:**

The distribution of test data and the demonstration examples is straightforward to understand and analyze. However, unlike traditional kernel regression, in-context learning necessitates a deeper comprehension of the relationship between pre-trained models and the input demonstrations. Specifically, it is important to understand how the kernel regression hypothesis explains the influence of pre-trained models on the similarity between $x_i$ and $x_{test}$, as well as the label space of $y_i$. In Section 4.2, the authors did not address this specific point. I am curious to know if the Kernel Regression hypothesis can provide any insights in understanding this aspect.

---

> ### Author Response · Authors · 2023-11-23
> **Author Response**
>
> We are very happy that the reviewer finds the investigated problem “timely and important” and the explanation “insightful”. We also thank the reviewer for the inspiring comments.
>
> **typo & grammar error**
>
> Thanks for pointing out the typos. We corrected it in the submission.
>
>
> **Q1: Relationship between pre-trained models and demonstrations**
>
> We agree that this is an important point where our explanation can provide insights. Intuitively, Theorem 1 implies a prediction output on $\mathbf{x}_{test}$ that is closer to labels $y_i$ of similar $\mathbf{x}_i$, and such similarity is related to how similar their conditional probabilities $P(\cdot | \mathbf{x})$are in the pre-trained LLM.
>
> To be specific, in Equation (4) the similarity kernel measures similarity between $\mathbf{x}$ and $\mathbf{x}’$ on the space of $\text{vec}(T_{\mathbf{x}})$ and $\text{vec}(T_{\mathbf{x}’})$. This is a flattened vector of the matrix $T_{\mathbf{x}}$, which defines the “belief state” in HMMs. The belief state determines the conditional probability $P(\cdot | \mathbf{x})$ after prefix $\mathbf{x}$. This is exactly the pre-training objective and functionality of LLMs. Therefore, if $x_i$ and $x_{test}$ have similar conditional probabilities (i.e., similar follow-up generations), their similarity kernel will have a larger value and the predicted label will be more similar to $y_i$, and vice versa. We add this part to Section 4.1.

---

### Official Review · Reviewer_kMXk · 2023-11-04

**Soundness:** 3 good
**Presentation:** 2 fair
**Contribution:** 3 good
**Rating:** 6
**Confidence:** 4

**Summary:**

The authors try to connect kernel regression with in-context learning (ICL) and explain some of its phenomenology established in prior work. Unlike prior works on the topic that often rely on synthetic setups, e.g., linear regression problems, the authors try to focus their experimental study on realistic/practical setups with open source LLMs and standard NLP datasets. That said, the theoretical framework used for hypothesis generation is grounded in a Hidden Markov model, as originally conceived by Xie et al., 2021.

**Strengths:**

The relation with kernel regression, in a sense, is a more formal take on the several recent works casting ICL as task selection (often such works also focus on real LLM setups). Even though I'm skeptical of this framework's ability to explain ICL in its entirety, and as the authors find it does at times fail to do so, I definitely find this perspective useful in the sense that it provides for provable/refutable hypothesis generation---such work is important for scientific development of black box systems like LLMs.

I also really like the experiments conducted in the paper: they are more mechanistic than prior task selection work in real LLMs, hypothesizing even a precise protocol for task selection, and more realistic than synthetic setups with linear regression or related tasks.

**Weaknesses:**

My primary apprehension with the paper is that the presentation is quite lacking. For example, in the theoretical model introduced in the main paper, the notion of observations is defined in regards to an HMM, but it is unclear if a datapoint $x$ is the same as these observations. In the proof in the appendix, though the algebra itself is accurate, somehow a datapoint $x$ is connected to the observations now and the HMM operates on them---this was rather unclear to me.

Similar to the above, the figures are often quite confusing. I can gather the primary takeaway based upon the experimental protocol, but the captions are very unclear and often axes are missing labels (e.g., in Figures 4/5). At times the figure itself is confusing as well: for example, in Figure 2, it is unclear to me how the attention maps have been plotted. The tokens in $x_{test}$ attend to prior tokens, but what is the y-axis per layer? I can guess it's the number of heads, but this should not be left to guess work.

Overall, I'm currently rejecting the paper despite having a position perspective on its contributions. Since ICLR allows changing the paper during rebuttals, if the presentation issues above can be addressed, I'm happy to update my score towards an accept.

A minor weakness/nitpick I want to point is that the paper often over-emphasizes, in my opinion, its theoretical contribution. I think the theoretical connection between ICL and kernel regression, for which the authors make asymptoticity assumptions, is best seen as a model to predict ICL's behavior. It would help to de-emphasize the theoretical contribution by primarily casting the paper's narrative as a model for *predicting*, not *explaining*, ICL's behaviors. As the authors note, the model does break at points (e.g., see page 6, remaining challenges). This implies claiming the model as an "explanation" is too strong.


**Post rebuttals comment:** Thank you for the response. Reading through the rebuttals above, discussion in other reviewers' comments, and rebuttals to them, I'm updating my score to a 6. I think the paper is well rounded and provides a nice predictive model for ICL. As another reviewer also noted however, the emphasis in the paper must reduce on "Explaining ICL"---the theory is making strict assumptions and is unable to explain all phenomenology. I'm raising the score hoping the authors will ensure making these changes.

**Questions:**

I'm curious as to what the authors think of related work on ICL that casts it as an optimization algorithm, such as (P)GD, over a model's layers. Such works are arguably taking a different ground than task selection / similarity search, as this paper does. Would you deem these works incorrect or, if not, then how do you think the two perspectives can be bridged?

---

> ### Author Response · Authors · 2023-11-23
> **Author Response**
>
> We appreciate the constructive and inspiring comments. We are happy that the reviewer finds our investigation perspective “useful” and our experiments “realistic”, and sees the overall direction “important” for LLMs.
>
> **W1: The sampling of $x$**
>
> The examples $[\mathbf{x}, y]$ are the same as the observations. About the sampling process, Section 3.2 explains that all demonstrative examples are i.i.d. sampled from the test task $\theta^\star$.
>
> **W2: Images and Captions**
>
> Thanks for pointing out places to improve the representation of the figures. Figure 4 has the same axes as Figure 3, where the x-axis is layers, and the y-axis denotes heads in each layer. We update the captions in the revision.
>
> In Figure 5, in the caption, we explain that the x-axis (0∼27) is layer number, the y-axis denotes the relative position to the high-attention position within each demonstration, and the z-axis is accuracy. We update the figure to make it clearer. In Figure 2, the y-axis per layer denotes the heads in the layers, which were annotated in the upper-right corner of Figure 2.
>
>
> **W3: theoretical contribution**
>
> The reviewer raises an important suggestion on the presentation of the paper. We agree that “explaining” might be a strong description of the contribution. Inspired by the reviewers’ suggestions, we propose to state that the work is “Understanding In-Context Learning from A Kernel Regression Perspective”, which avoids the problems in “explanation” but also takes our computational feature analysis into consideration. In the revision, we updated the abstract and introduction. We also plan to revise the title at the camera-ready phase, as ICLR now forbids title updates during rebuttal.
>
> **Q1: GD-based explanations**
>
> The reviewer raises a very interesting question on the relation with other explanations. We would answer that these two perspectives do not conflict with each other.
>
>  First, currently, two explanations account for different scenarios of ICL: our explanation focuses on LLMs on linguistic data, while most (P)GD-based explanations are constructed on synthetic 1-length datasets, which have very different data distribution and format. Therefore, two explanations can be viewed as complementary perspectives in different cases.
>
>  Secondly, two explanations might be remotely connected in the future. One such intuition can be found in the neural tangent kernel (NTK) (Jacot et al., 2018, https://proceedings.neurips.cc/paper/2018/hash/5a4be1fa34e62bb8a6ec6b91d2462f5a-Abstract.html) field, which builds a connection between gradient descent optimization and kernel regression predictions. Such a connection is still weak, as NTK builds on strong assumptions about a wide enough network and particular architecture. But this still points to a promising direction for bridging the gap.
>
> Finally, we hope that this addresses your concerns. If you find our response helpful in answering your questions, we kindly request you to consider updating the reviews.

---

### Official Review · Reviewer_Hu7i · 2023-11-06

**Soundness:** 2 fair
**Presentation:** 2 fair
**Contribution:** 2 fair
**Rating:** 5
**Confidence:** 4

**Summary:**

The authors derive that In-Context Learning in LLMs can be viewed as kernel regression and also explores some practical investigation to support this idea.

**Strengths:**

The paper tries to establish/proof that ICL can be seen as kernel regression and conducts some practical investigation to back the theoretical claims.

**Weaknesses:**

Theoretical: The authors primary assumption in Section 2.1 states that “as the number of samples n increase ...  converges to a
kernel-regression for”. The reviewer thinks this is a very strong assumption since this begs two questions (i) How does ICL work so well for only 2-3 examples especially for large models and (ii) Why does ICL performance saturate usually after 10-15 examples?

Practical: The practical investigation in this paper is weak. While the investigations of the attention layers are interesting, but they are neither enough to back the theoretical findings nor show practical benefits of the findings. The reviewer liked Section 4.2 and would have liked to seen these explored practically in the paper.

**Questions:**

Theoretical: The authors primary assumption in Section 2.1 states that “as the number of samples n increase ...  converges to a
kernel-regression for”. The reviewer thinks this is a very strong assumption since this begs two questions (i) How does ICL work so well for only 2-3 examples especially for large models and (ii) Why does ICL performance saturate usually after 10-15 examples?

Practical: The practical investigation in this paper is weak. While the investigations of the attention layers are interesting, but they are neither enough to back the theoretical findings nor show practical benefits of the findings. The reviewer liked Section 4.2 and would have liked to seen these explored practically in the paper.

I would love the authors' responses on the limitations listed above.

---

> ### Author Response · Authors · 2023-11-23
> **Author Response**
>
> We thank the reviewer for the time and review and are pleased to see that they like the insights in Section 4.2. Regarding the comments and questions:
>
> **W1 & Q1: Understanding the main theorem**
>
> The reviewer raises an interesting concern. If we understand correctly, the statement appears in Section 4.1 instead of Section 2.1. Besides, it is an explanation of the proven Theorem 1 results instead of an assumption. As for the fact that LLMs work well for only 2\~3 examples and saturate after 10\~15 examples, we would like to point out that this phenomenon is a different problem from Theorem 1. The mentioned phenomenon is about “how ICL outputs converge to ground truth,” while Theorem 1 studies “how the kernel-regression interpretation converges to ICL outputs”. Therefore, it does not conflict with Theorem 1.
>
> **W2 & Q2: Practical investigation**
>
> We thank the reviewer for the constructive comment. Limited by rebuttal period, we conduct an analysis to verify the insight that OOD examples affect ICL performance. Specifically, on the SST2 dataset, we prompt GPT-3.5-turbo to generate semantically similar while rarer (i.e. OOD) expressions of inputs and use them to substitute the original dataset inputs. We consider 3 types of OOD types. “Rare word” is where words are substituted with rare synonyms. “Complex” is to express the original sentence in a more complex structure. “Typo” is where we require adding typos to the original input. The results are listed below. We see that these OOD types more or less decrease the ICL accuracy, with “typo” having the largest effect, dropping the accuracy to near random. This accords with our insight in Section 4.2 that OOD samples can negatively affect ICL performance. We add these results in the revision.
>
> | OOD-type | None | rare word | complex | typo |
> | --- | --- | --- | --- | --- |
> | accuracy | 0.805 | 0.677 | 0.788 | 0.534 |
>
> We hope this addresses your concerns and hope that you could kindly consider updating the review scores.

---

### Official Review · Reviewer_Gjpv · 2023-11-06

**Soundness:** 3 good
**Presentation:** 2 fair
**Contribution:** 2 fair
**Rating:** 5
**Confidence:** 3

**Summary:**

This paper tackles the challenge of elucidating the mechanism enabling in-context learning in LLMs and its being acquired through autoregressive pretraining. The hypothesis put forth in the paper is that Transformer-based LLMs pretrained in this way acquire the capability to perform in-context learning by performing a kernel regression operation based on the demonstration present in the context window. The paper analyses this hypothesis formally by modeling pretraining over text sequences as learning an HHM within the OOM formalism. This allows to formally relate the in-context prediction to the HHM modeling the pretraining dataset.

**Strengths:**

* Interesting and novel approach to the problem of explaining in-context learning in LLMs using an intriguing modeling approach (OOM formalism to model HHMs).
* The main hypothesis put forth by the paper is crisply stated and clear to understand.

**Weaknesses:**

* Connection with OOM formalism is intriguing but not explored in what seems like it's full quantitative potential to provide practical guidance for LLMs pretraining or prompt engineering.
* As mentioned also in the paper, the kernel regression formalism does not capture some crucial phenomenology observed in in-context learning, like the sensitivity to in-context samples order, and there are no indications provided in the paper of how this shortcoming in the formalism could be addressed moving forward.
* The notation developed in the paper mostly results in an analysis (based on the heuristic similarity between QKV-attention and kernel regression) of the last tranformer layer.
* Formatting and notation of the article is somehow problematic with some symbols not being defined by the time they are used in the text. This makes understanding the details of the paper potentially tough, diminishing its impact. For instance, $\Sigma_{p_{pre-train}}$ appearing below eq (3) in the definition of $\epsilon_\theta$ is not defined.

**Questions:**

* Is this specific to transformers? How about LMs based on RNNs or attention-free architectures?
* Did the authors test a toy benchmark where the pretraining sequences are generated from a known HHM where one can control size of the pretraining datasets and complexity of the HHM to empirically verify the convergence in Theorem 1? This seems like it would be a great way to verify the main theory result.
* Theorem 1  mentions that triangular brackets indicate the inner product, but this notation is not used in the theorem.
* Eq (7): sum over j but the index used is i.

---

> ### Author Response · Authors · 2023-11-23
> **Author Response**
>
> We thank the reviewer for the review and time. We are glad to see that the reviewer finds the proposed approach “intriguing,” “Interesting and novel,” and the hypothesis “crisply stated and clear to understand”. Our response to your comments is as follows, which we also adopted in the paper revision:
>
> **W1: OOMs**
>
> We agree with the reviewer other algorithms in OOMs could potentially be useful in partially explaining LLMs. Still, in this paper, we only need the forward prediction algorithm during the inference stage (because the setting of ICL is inference only, no learning involved), so many techniques in OOMs regarding the learning stage will not be used. Furthermore, we kindly note that OOMs share many similarities with HMMs, so the analysis results by using OOMs should be similar to the current status.
>
> **W2: Overcoming the Limitations**
> We agree that these challenges are important to deal with in future work. Answers to these questions require a deeper analysis of the theoretical structure of linguistic data and their mathematical characterization. Specifically, future work should explore a theoretical framework more suitable for studying natural language, which is beyond the capability of conventional tools like HMMs and mixtures of distributions.
>
> **W3: Application to Other Layers**
>
> We thank the reviewer for this interesting question. Our explanation does not specify or limit the model from implementing kernel regression in a certain layer. In fact, this kernel regression can be implemented in one or more layers as long as these (possibly redundant) results can be passed to the final layer and aggregated. This “passing” can be easily implemented by the skip connections. We add this explanation to Section 5.1
>
> **W4: Notations**
>
> Thank you for suggesting improvements on notations. The covariance matrix $\Sigma_{p_\text{pre-train}}$ was a typo and supposed to be $\Sigma_{p_\text{pre-train}, l}$, which is defined in the last paragraph of Section 3.2 to be the covariance of elements in $T_{\mathbf{o}}$, when the sequence $\mathbf{o}$ is sampled from the initial distribution $p_\text{pre-train}$. In the same paragraph, $\epsilon_\theta$ is defined as $\inf_l \rho(\Sigma_{p_\text{pre-train}, l}^{-1} - \Sigma_{s_\theta, l}^{-1})$, where $s_\theta$ is the task-specific initial state, and $\rho$ is the spectral radius of a matrix. This quantifies the difference between pre-training and task-specific distributions. We also updated the revision accordingly.
>
> **Q1: Specificity to Transformers**
>
> Our explanation provides a mechanism of how attention-based models, such as Transformers and attention-RNNs, might be tackling the ICL problem. Other architecture might process the ICL inputs differently.
>
> **Q2: A Synthetic HMM**
>
> Thanks for the constructive suggestion, and we agree that this helps the clarity and convincingness of the analysis. We experimented on randomly parameterized synthetic HMMs with 8 tasks, 80 states, and 100 observations. We vary the number of samples and evaluate the proposed kernel regression on fitting the Bayesian posterior, which is the intuition in Theorem 1. Results are listed in the table below. We indeed see a decreasing loss and increasing accuracy with more demonstrative examples. The loss also converges to a non-zero value according to Equation 6. These results are added to Appendix C.
>
> | #samples | 1 | 2 | 4 | 8 | 16 | 32 | 64 | 128 |
> | --- | --- | --- | --- | --- | --- | --- | --- | --- |
> | distance | 1.322 | 0.942 | 0.519 | 0.218 | 0.163 | 0.085 | 0.093 | 0.083 |
> | accuracy | 0.337 | 0.532 | 0.745 | 0.901 | 0.926 | 0.968 | 0.961 | 0.964 |
>
>
> **Q3 & 4: Notations**
>
> Thanks for pointing out the typos. The inner product in Eq (4) is an unused notation. The summation notation in Eq (7) also contains a typo. We corrected them in the revision.
>
> Finally, we hope that this addresses your concerns. If you find our response helpful in answering your questions, we kindly request you to consider updating the reviews.

---

> > ### Comment · Reviewer_Gjpv · 2023-11-23
> >
> > Thanks for engaging with my feedback. The synthetic experiments mentioned in the rebuttals sound very interesting.
> > As for my other points, my comments about OOM was not a criticism for using that formalism instead of HHM, quite the opposite. OOMs are a strict generalization of HHMs, and seem very well suited for analyzing your hypothesis, as your approximation result in Theorem 1 also nicely shows. My issue (and in general my main criticism) was mostly that this interesting result seems under-exploited as it only leads to a heuristic correspondence between kernel regression and the operation of the last self-attention layer. In particular, the question was whether this formalism could be pushed a bit further to explain the rest of the architecture and its role in learning and inference, and maybe even provide practical guidance that would impact how pretraining or prompt-engineering are done in practice.

---

### Author Response · Authors · 2023-11-23
**General Response and Revision Summary**

We thank all reviewers for their constructive and inspiring reviews. We are glad to see that reviewers find that the approached problem of understanding language model in-context learning is “important” (reviewers kMXk and oa7R), and the proposed perspective is ”interesting” (reviewer Gjpv, Hu7i) and “novel” (Gjpv and iW9p).

In this work, we offer a kernel-regression-based perspective for understanding ICL. We empirically verify the analogy between kernel regression and the internal representations of LLMs on ICL inputs. This understanding perspective also offers ways to interpret a few observations, such as the selection of similar and in-distribution samples.

During rebuttal, we address the reviewers’ concerns and revise the submission document accordingly. Some key additions to the paper include:

**Synthetic HMM** (Gjpv) We experiment on a synthetic HMM to intuitively verify the Theorem 1.

**Analyzing the Selection of In-Distribution Samples** (Hu7i & iW9p) We conduct an experiment for confirming the insight offered by the kernel-regression perspective that in-distribution samples benefit ICL performance.

**Effect of the Pre-trained LM** (oa7R) We discuss the effect of pre-trained LM on the kernel.

**Figure Representation** (kMXk, iW9p) We update the figures and captions for better clarity.

**Paper Framing** (kMXk, iW9p) We revise the statement in the paper to more accurately and clearly express the messages it conveys.

**Llama-2 Model** (iW9p) We also visualize the attentions on Llama-2.

These additions aim to augment the paper and to improve the understanding of the proposed perspective for understanding ICL. We thank all reviewers again for the valuable time and reviews.

---

### Meta-Review · Area_Chair_Shyj · 2023-12-08

**Metareview:**

This paper sets out to explain in-context learning as kernel regression. The paper derives a theorem that supports this claim under several strong assumptions, followed by several experiments that demonstrate that the attention and hidden features in LLMs during ICL exhibit correlations similar to kernel regression.

While the reviewers and myself think the empirical findings are interesting, the framing of the paper suffers from serious issues.

- Kernel regression is agnostic to the order at which the examples are fed, whereas ICL is highly dependent on the order of the examples. This alone rejects the hypothesis that ICL could be explained as kernel regression.

As such, I think the only logical way is to use kernel regression as a tool to shed light on ICL instead of claiming that ICL is kernel regression. In doing so, the authors are encouraged to study some of the intriguing phenomena associated with ICL. For example, here are some of the shortcomings that could be improved:

1. The intriguing phenomena with ICL arise from the way examples are selected. This is not investigated in this paper.

2. The assumptions used in the theoretical derivations are too unrealistic. So the theoretical result probably needs to be dropped as the hypothesis that ICL is kernel regression is evidently false.

3. The performance of ICL highly depends on the design of prompts, which is unexplained by this work.

4. One of the intriguing phenomena of ICL is generalization to labels outside of the in-context exemplar. That remains totally unexplained by this framing.

5. The authors are encouraged to investigate how a simple naive kernel would fare against ICL, or how the findings of these paper could be used to find a better kernel to compare ICL directly with kernel regression.

6. The empirical results should be reframed as understanding of ICL, as they do not actually verify the main hypothesis of the paper, as these phenomena could be explained in other simpler ways.

Given this, the paper is recommended to be rejected at this time, and we hope the authors can use this feedback to further develop their work as probing ICL rather than claiming ICL = kernel regression.

**Justification For Why Not Higher Score:**

The paper suffers from a serious framing flaw. Unfortunately, it is evident that the main hypothesis that is proposed by this paper (namely ICL is kernal regression) is false. Based on this fact alone, we cannot recommend to accept the paper.

**Justification For Why Not Lower Score:**

The paper has interesting empirical findings that basically use kernel regression framing to correlate the decisions of a large model. These findings if developed further can become a publishable paper.

---

### Decision · Program_Chairs · 2024-01-16

Reject